

# Airborne wind lidar observations over the North Atlantic in 2016 for the pre-launch validation of the satellite mission Aeolus

Oliver Lux, Christian Lemmerz, Fabian Weiler, Uwe Marksteiner, Benjamin Witschas, Stephan Rahm, Andreas Schäfler, Oliver Reitebuch

5   Deutsches Zentrum für Luft- und Raumfahrt e.V. (DLR), Institut für Physik der Atmosphäre, Oberpfaffenhofen 82234, Germany

*Correspondence to*: Oliver Lux (oliver.lux@dlr.de)

**Abstract.** In preparation of the satellite mission Aeolus carried out by the European Space Agency, airborne wind lidar observations have been performed in the frame of the North Atlantic Waveguide and Downstream Impact Experiment 10   (NAWDEX), employing the prototype of the satellite instrument, the ALADIN Airborne Demonstrator (A2D). The direct-detection Doppler wind lidar system is composed of a frequency-stabilised Nd:YAG laser operating at 355 nm, a Cassegrain telescope and a dual-channel receiver. The latter incorporates a Fizeau interferometer and two sequential Fabry-Pérot interferometers to measure line-of-sight (LOS) wind speeds by analysing both Mie and Rayleigh backscatter signals. The benefit of the complementary design is demonstrated by airborne observations of strong wind shear related to the jet stream 15   over the North Atlantic on 27 September and 4 October 2016, yielding high data coverage in diverse atmospheric conditions. The paper also highlights the relevance of accurate ground detection for the Rayleigh and Mie response calibration and wind retrieval. Using a detection scheme developed for the NAWDEX campaign, the obtained ground return signals are exploited for the correction of systematic wind errors. Validation of the instrument performance and retrieval algorithms was conducted by comparison with DLR's coherent wind lidar which was operated in parallel, showing a systematic error of the 20   A2D LOS winds of less than 0.5 m·s$^{-1}$ and random errors from 1.5 m·s$^{-1}$ (Mie) to 2.7 m·s$^{-1}$ (Rayleigh).

## 1 Introduction

Over the last decade, Doppler wind lidar systems (Reitebuch, 2012a) have emerged as a versatile tool for the range-resolved detection of wind shears (Shangguan et al., 2017), aircraft wake vortices (Köpp et al., 2004; Dolfi-Bouteyre et al., 2009), wind and temperature turbulence (Banakh et al., 2014) as well as gravity waves (Witschas et al., 2017), amongst other 25   applications. In particular, direct-detection wind lidars have been demonstrated to provide accurate wind information from ground up to altitudes of 60 km (Dou et al., 2014) or even beyond (Baumgarten, 2010; Hildebrand et al., 2012). The most ambitious endeavour in this context is the upcoming satellite mission Aeolus of the European Space Agency (ESA) which strives for the continuous, global observation of atmospheric wind profiles employing the first ever satellite-borne Doppler wind lidar instrument ALADIN (Atmospheric LAser Doppler INstrument) (ESA, 2008; Stoffelen et al., 2005). Being a part





of ESA's Living Planet Programme, Aeolus will significantly contribute to the improvement in numerical weather prediction (NWP), as it will close the gaps in the wind data coverage, especially over the oceans, which has been identified as one of the major deficiencies in the current Global Observing System (Baker et al., 2014; Andersson, 2016). For this purpose, it will provide one line-of-sight (LOS) component of the horizontal wind vector from ground throughout the troposphere up to the

lower stratosphere (about 27 km) with a vertical resolution of 0.25 km to 2 km depending on altitude and precision of 1 m·s⁻¹ to 3 m·s⁻¹ (ESA, 2016; Reitebuch, 2012b). The obtained data will allow for greater accuracy of the initial atmospheric state in NWP models and thus improve the quality of weather forecasts (Tan and Andersson, 2005) as well as the understanding of atmospheric dynamics and climate processes (ESA, 2008). As a secondary product, the wind lidar system, which is scheduled for launch in 2018, will provide information on cloud top heights and on the vertical distribution of clouds and

aerosol properties such as backscatter and extinction coefficients (Flamant et al., 2008; Ansmann et al., 2007).

Over the past years, a prototype of the Aeolus payload, the ALADIN Airborne Demonstrator (A2D), has been developed and deployed in several field experiments, aiming at pre-launch validation of the satellite instrument and at performing wind lidar observations under various atmospheric conditions (Reitebuch et al., 2009; Marksteiner et al., 2009; Marksteiner et al., 2017). Most recently, in autumn of 2016, the A2D was employed in the frame of the North Atlantic Waveguide and

Downstream Impact Experiment (NAWDEX) (Schäfler et al., 2018). Based in Keflavík, Iceland, this international field campaign had the overarching goal to investigate the influence of diabatic processes, related to clouds and radiation, on the evolution of the North Atlantic jet stream. Accurate wind speed observations of the North Atlantic jet stream form the basis for quantifying effects of disturbances for downstream propagation and related high-impact weather in Europe. For this purpose, four research aircraft equipped with diverse payloads were employed which allowed for the observation of a large

set of atmospheric parameters using a multitude of state-of-the art remote sensing instruments, while ground stations delivered a comprehensive suite of additional measurements to complement the meteorological analysis.

With a view to the forthcoming Aeolus mission, the NAWDEX campaign was an ideal platform for extending the wind dataset obtained with the A2D, as it offered the opportunity to perform wind measurements in dynamically complex scenes, including strong wind shear and varying cloud conditions. Furthermore, multiple instrument calibrations, which are a

prerequisite for accurate wind retrieval, could be conducted over ice, namely the Vatnajokull glacier on Iceland, ensuring high signal-to-noise ratios (SNR) of the ground return and thus low systematic errors. In addition, the large-scale cooperation of atmospheric research groups from around the world was beneficial for the preparation of the upcoming launch of Aeolus.

Among the 14 research flights conducted in the frame of NAWDEX, the two flights performed on 27 September and on 4 October 2016 were especially interesting with regard to the instrument-driven goals of the campaign. While the former flight

was characterized by exceptionally high wind speeds and strong wind shear to be sampled by the A2D, the latter one provided ground visibility which allowed for the analysis of ground return signals. In general, analysis of the ground return offers many possibilities for improving the performance of lidar instruments. Recently, Amediek and Wirth (2017) introduced a method for quantifying laser pointing uncertainties in airborne and spaceborne lidar instruments which is based on the comparison of ground elevations derived from the lidar ranging data with elevation data from a high-resolution digital



elevation model. Regarding airborne wind lidar and radar systems, ground echoes can be exploited to account for systematic pointing errors and to determine the mounting angles of the instrument. Here, the ground surface is used as a zero wind reference which allows to estimate the contribution of the aircraft motion to the actual atmospheric wind measurement, and hence to correct for inaccuracies in the aircraft attitude data as well as in the instrument's alignment (Bosart et al., 2002;

Kavaya et al., 2002; Chouza et al., 2016a; Weiler, 2017). Accurate zero wind correction (ZWC), however, requires precise differentiation between atmospheric and ground return signals in order to prevent systematic errors. This is particularly true for the A2D (and ALADIN) due to its coarse vertical resolution of several hundred meters. Hence, in contrast to previous A2D airborne campaigns, an enhanced scheme for the detection of ground return signals was developed for NAWDEX.

The paper is organized as follows. First, the operation principle of the system is described with a focus on the

complementary design of the instrument comprising two different receiver channels which allow for the analysis of both particle and molecular backscatter signals. The subsequent section is devoted to the Rayleigh and Mie response calibrations which represent an essential part of the data analysis. Here, the implemented ground detection method used for the A2D data analysis is introduced. Comparison with the approach taken in previous campaigns reveals the influence of the surface albedo on the quality of Rayleigh and Mie response calibrations and highlights the necessity of proper ground detection.

Afterwards, wind observations performed with the A2D during the two above-mentioned NAWDEX flights are presented, demonstrating the ability of the lidar system to provide wind profiles with broad data coverage under various atmospheric conditions. Evaluation of the data accuracy and precision is conducted by comparing the measured wind speeds with those obtained by DLR's coherent wind lidar system (Weissmann et al., 2005; Witschas et al., 2017) which was operated in parallel from the same aircraft as a reference system. Finally, ZWC based on the refined ground detection scheme is shown

to provide a significant reduction of the systematic wind error for the second flight.

## 2 The A2D direct-detection wind lidar system

The A2D wind lidar is composed of a pulsed, frequency-stable, ultra-violet (UV) laser transmitter incorporating a reference laser system, a Cassegrain telescope, a configuration of optical elements (front optics) to spatially overlap a small portion of the outgoing radiation with the return signals from the atmosphere and the ground, and a dual-channel receiver including

detectors. A schematic of the lidar is depicted in Fig. 1. The individual components will be described in the following.

### 2.1 Laser transmitter, telescope and front optics

The laser transmitter of the A2D is based on a frequency-tripled Nd:YAG master oscillator power amplifier (MOPA) system, generating 20 ns pulses (full width at half maximum (FWHM)) at 354.89 nm wavelength. The injection-seeded laser which uses an active frequency stabilization technique provides single-frequency UV pulses with energy of 60 mJ at 50 Hz

repetition rate (3.0 W average power), while showing near-diffraction-limited beam quality. A comprehensive description of the laser transmitter configuration and its performance is provided in Lemmerz et al. (2017) and Schröder et al. (2007).



In the last years, particular attention has been devoted to the cavity control mechanism which ensures high single frequency operation stability even under vibration conditions. In addition to the strict requirements in terms of frequency stability, a further challenge is imposed by the necessity to trigger the receiver electronics about 60 µs before the laser pulse emission with an error of less than 100 ns. Therefore, a dedicated active frequency stabilization technique was developed which is based on the Ramp-Delay-Fire method (Nicklaus et al., 2007). Fast detection of the master oscillator cavity resonances with the seed laser frequency enabled effective compensation of higher-frequency vibrations, while providing a sufficiently early trigger for the detector electronics with a timing stability of around 80 ns (Lemmerz et al., 2017).

Measurement of the transmitted laser frequency and calibration of the frequency-dependent transmission of the receiver spectrometers are prerequisite for accurate wind retrieval. Therefore, a small portion of the pulsed UV laser radiation, referred to as internal reference, is collected by an integrating sphere, coupled into a multimode fiber (200 µm core diameter) and guided to the receiver via the front optics, while allowing adjustable signal levels by using a variable fiber attenuator (see laser transmitter in Fig. 1). Another small fraction of the beam is directed to a wavelength meter (HighFinesse, WS Ultimate 2) with a relative accuracy of $10^{-8}$ in order to monitor the UV frequency of the outgoing laser pulse.

The spatial properties of the high-energy laser were characterised prior to the NAWDEX campaign according to the ISO 11146 standard (ISO, 2005), yielding a beam quality factor ($M^2$) of 1.1 for both the major and minor beam axis. As a result, after passing through the beam expander, the collimated beam showed a full-angle divergence ($\pm 3\sigma$, containing >99% of the energy) of 98 µrad and 102 µrad at $4\sigma$-beam diameters of 7.3 mm and 7.1 mm for the two axes.

The UV laser is transmitted into the atmosphere via a piezo-electrically controlled mirror that is attached to the frame of a Cassegrain-type telescope, as shown in Fig. 1. In contrast to ALADIN that incorporates a 1.5 m diameter telescope and will operate at an off-nadir pointing angle of 35°, the A2D employs a 0.2 m telescope which is oriented at an off-nadir angle of 20°. The convex spherical secondary mirror of the telescope collects the backscattered light and guides it to the front optics of the A2D receiver assembly. The structural design of the telescope causes a range-dependent overlap function which has to be considered in the wind retrieval as it reduces the backscatter signal (Paffrath et al., 2009).

Aside from a narrowband UV bandpass filter (FWHM: 1.0 nm) which blocks the broadband solar background spectrum, the front optics include an electro-optic modulator (EOM) for temporal separation of the atmospheric signal from the internal reference signal. The latter is injected into the front optics assembly via the aforementioned multimode fiber, so that both signals enter the spectrometer optics on equal paths. In addition, active stabilization of the laser beam pointing is realised by a co-alignment control loop. For this purpose, a portion of the backscattered signal passing through the front optics is imaged onto a UV camera (SONY XC-EU50CE) to monitor the horizontal and vertical position of the centre of gravity (CoG) of the beam. A reference position ($CoG_X/CoG_Y$) is defined and a feedback loop involving three piezo-actuators mounted on the last laser transmit mirror is applied to actively stabilise the co-alignment of the transmit and receive path of the laser beam. In this way, variations in the incidence angle of the atmospheric return signals on the receiver spectrometers are reduced. This is crucial for accurate wind measurements, especially for the Rayleigh channel, as angular variations of 1 µrad with respect to the 200 mm telescope diameter and a field-of-view (FOV) of 100 µrad introduce errors of the horizontal wind speeds of



up to 0.4 m·s$^{-1}$, as derived from optical simulations and experiments (DLR, 2016). It should be noted that active stabilization of the transmit-receive co-alignment is not required for the satellite instrument, since the same telescope is used for transmission of the laser beam and reception of the backscattered signals.

## 2.2 Dual-channel receiver and detectors

The receiver optics of both the satellite instrument and the A2D are almost identical and consist of two different spectrometers, as shown on the right-hand side of Fig. 1. Two sequential Fabry-Pérot interferometers (FPIs) are employed for measuring the Doppler frequency shift of the broadband Rayleigh backscatter signal from molecules, whereas a Fizeau interferometer is used for determining the Doppler shift of the narrowband Mie signal originating from cloud and aerosol backscattering. Detection of the two signals is realized by using two accumulation charge-coupled devices (ACCDs) which

allow for data acquisition in 24 range gates, where the vertical resolution within one profile can be varied from 296 m to about 2 km.

The wind measurement principle of the A2D wind lidar system is based on detecting frequency differences between the emitted and the backscattered laser pulses. Due to the Doppler effect, the frequency $f_0$ of the outgoing pulse is shifted upon backscattering from moving particles (cloud particles, aerosols, molecules). The frequency shift in the backscattered signal

$\Delta f_{\text{Doppler}}$ is proportional to the wind speed $v_{\text{LOS}}$ along the laser beam LOS: $\Delta f_{\text{Doppler}} = 2\, f_0/c \cdot v_{\text{LOS}}$, with $c$ being the speed of light. For an emission frequency of $f_0 = 844.75$ THz (354.89 nm vacuum wavelength), a LOS wind speed of 1 m·s$^{-1}$ translates to a frequency shift of 5.63 MHz which corresponds to a wavelength shift of 2.37 fm. The required accuracy of the frequency measurement is hence in the order of $10^{-8}$ to measure wind speeds with an accuracy of 1 m·s$^{-1}$. Owing to the large difference in spectral width of the Mie (~50 MHz) and Rayleigh (~3.8 GHz at 355 nm and 293 K) atmospheric backscatter

signals, two different techniques are applied for deriving the Doppler frequency shift from the two spectral contributions separately.

The measurement principle of the Rayleigh channel relies on the double-edge technique (Chanin et al., 1989; Garnier and Chanin, 1992; Flesia and Korb, 1999; Gentry et al., 2000) and involves two bandpass filters (A and B) which are placed symmetrically around the frequency of the emitted laser pulse, as illustrated in Fig. 2(a). The width and spacing of the filter

transmission curves (free spectral range (FSR): 10.95 GHz, FWHM: 1.78 GHz, spacing: 6.18 GHz) is chosen such that the maxima are close to the inflexion points (edges) of the molecular line that is spectrally broadened by virtue of Rayleigh-Brillouin scattering (Witschas, 2011a; Witschas, 2011b; Witschas, 2011c). The transmitted signal through each filter is proportional to the convolution of the respective filter transmission function and the line shape function of the atmospheric backscatter signal. Consequently, the contrast between the return signals $I_A$ and $I_B$ transmitted through filters A and B

represents a measure of the frequency shift between the emitted and backscattered laser pulse, thus defining the frequency-dependent Rayleigh response $\Psi_{\text{Ray}}$ as follows:

$$\Psi_{\text{Ray}}(f) = \frac{I_A(f) - I_B(f)}{I_A(f) + I_B(f)}. \tag{1}$$





Close to the filter cross point, where the transmission functions intersect, the relationship between Rayleigh response and frequency is approximately linear with a slope of about $5 \cdot 10^{-4}$ MHz$^{-1}$.

The determination of the Doppler shift from the narrowband Mie return signal is based on the fringe-imaging technique (McKay, 2002) involving the measurement of the spatial location of an interference pattern, as shown in Fig. 2(b). For this

purpose, a Fizeau interferometer is used consisting of two plane-parallel plates that are tilted by a small angle of several μrad. Due to the wedge angle, the linear interference pattern (fringe) is produced at a distinct lateral position along the wedge where the condition for constructive interference is fulfilled. Hence, a Doppler frequency shift of the signal results in a spatial displacement of the fringe which is vertically imaged onto the ACCD detector, whereby the relationship between the Doppler shift and the centroid position of the fringe $x$ is approximately linear ($\Delta x \approx k \cdot \Delta f_{\text{Doppler}}$), so that the Mie response

reads:

$$\Psi_{\text{Mie}}(f) = x(f) = x(f_0) + \Delta x(f) = x_0 + k \cdot \Delta f_{\text{Doppler}}, \qquad (2)$$

The relative shift of the fringe $k$ is in the order of 100 MHz/pixel. From the Fizeau FSR of 2.2 GHz, only a section of 1.6 GHz is recorded by the 16 pixel columns of the ACCD (imaged spectral range), resulting in an effective LOS wind measurement range of ±145 m·s$^{-1}$.

The thinned and back-side illuminated ACCD with 16 x 16 pixels is optimised for operation in the UV showing a high quantum efficiency of 85%, while cooling to -30°C provides a low electronic noise level. The electronic charges generated in the imaging zone of the device are accumulated directly in a memory zone within the CCD chip, thus allowing for low readout noise (Reitebuch et al., 2009). For the ACCD used in the Mie channel, the electronic charges of all 16 rows are binned together to one row for each range gate of each laser pulse, resulting in 16 spectral channels of about 100 MHz width.

For the Rayleigh channel, the two spots produced by the two FPIs are imaged onto the left and right half of a second ACCD of the same type, with the centres of the spots being separated by 8 pixels (see bottom right part of Fig. 1). As for the Mie channel, the electronic charges of all 16 rows are binned together to one row, whereas the signal of each Rayleigh filter is contained in 6 pixels that are summed up in the retrieval algorithms after digitisation.

The memory zone of the ACCD contains 25 rows so that a maximum number of 25 range gates can be acquired, from which

three range gates are used for detecting the background light, the detection chain offset (DCO) and the internal reference signal, while two range gates act as buffers for the internal reference. The DCO is a constant electric voltage at the analogue-to-digital converter. The atmospheric backscatter signals are collected in the remaining 20 (so-called atmospheric) range gates. The transfer time from the image to the memory zone limits the minimum temporal resolution of one range gate to 2.1 μs, which corresponds to a range resolution of 315 m and a height resolution of 296 m, taking account of the 20°-off-

nadir pointing of the instrument. The timing sequences of both ACCDs are programmable, providing flexible and independent vertical resolution for the Rayleigh and Mie wind profiles.

The horizontal resolution of the A2D is determined by the acquisition time of the detection unit. Here, the signals obtained from 20 laser pulses are accumulated to so-called *measurements* (duration 0.4 s), while the combination of the signals from 35 measurements (700 pulses) constitutes one *observation* (duration 14 s). Considering the time required for data read out





and transfer (4 s), the separation time between two subsequent observations thus accounts for 18 s. For a typical ground speed of the Falcon aircraft of 200 m·s⁻¹, this results in a horizontal resolution of 3.6 km. Note that continuous data readout without gaps of 4 s is carried out for the satellite instrument on Aeolus, but the concept for on-chip averaging of multiple laser pulse returns to measurements is used as well. In the following, the terms *observation* and *measurement* are
consistently used referring to the sampling of the A2D data.

## 3 Response calibrations and ground detection

The A2D direct-detection wind lidar system was employed during the NAWDEX field experiment delivering valuable data with a view to the pre-launch activities for the upcoming Aeolus mission as well as with regards to the meteorological objectives of the campaign. In the framework of NAWDEX, 14 research flights have been performed with the Falcon aircraft
of DLR, including four transfer flights between Oberpfaffenhofen, Germany and the airbase in Keflavík, Iceland. An overview of the flights, wind scene periods and the number of A2D observations is presented in Table 1. 27 flight legs with continuous sampling of wind profiles were conducted with periods ranging from 11 minutes to more than one hour, adding up to almost 15 hours over the whole campaign. From the 14 research flights, two flights on 28 September and 15 October 2016 were dedicated to the calibration of the A2D instrument. This procedure represents a key part of the wind retrieval and
will be described in this chapter. Here, the focus is put on a ground detection scheme that allows for accurate identification of ground signals and hence reduced systematic errors of the calibration parameters.

### 3.1 Response calibrations

Spectral response calibration of the A2D is a prerequisite for the wind retrieval, since the relationship between the Doppler frequency shift of the backscattered light, i.e. the wind speed, and the response of the two spectrometers has to be known for
the wind retrieval. In particular, proper knowledge of the Rayleigh response for different altitudes is necessary, as the spectral shape of the Rayleigh-Brillouin backscatter signal significantly depends on temperature and pressure of the sampled atmospheric volume (Witschas et al., 2014) and thus varies along the laser beam path.

For deriving the frequency-dependence of the Rayleigh and Mie channel spectral response, a frequency scan of the laser transmitter is carried out, thus simulating well-defined Doppler shifts of the radiation backscattered from the atmosphere
within the limits of the laser frequency stability. During the calibration, the contribution of (real) wind related to molecular or particular motion along the instruments' LOS has to be eliminated, i.e. the LOS wind speed $v_{LOS}$ needs to be zero. In practice, this is accomplished by flying curves at a roll angle of the Falcon aircraft of 20°, resulting in approximate nadir pointing of the instrument and hence $v_{LOS} \approx 0$, while assuming that the vertical wind is negligible. Consequently, regions with expectable non-zero vertical winds, e.g. introduced by gravity waves or convection are avoided in this procedure. Nadir
pointing leads to a circular flight pattern of the aircraft which is preferably located over areas with high surface albedo in the UV spectral region (e.g. over ice), hence enabling strong ground return intensities and, in turn, high SNR. In the course of



the calibration procedure which takes about 24 minutes, highest attention has to be paid to the minimization of all unknown contributions to the Rayleigh and Mie response such as biases resulting from inaccurate co-alignment of the transmit and receive path, temperature variations of the spectrometers or frequency fluctuations of the laser transmitter.

During NAWDEX, six response calibrations have been carried out over Iceland, four over the Vatnajokull glacier and two over ice-free land in the north of the island. During each calibration, the laser frequency was tuned in steps of 26 MHz (corresponding to 4.5 m·s⁻¹) over a 1.4 GHz interval (±125 m·s⁻¹) and the Rayleigh and Mie responses were determined after averaging over 700 pulses (1 observation) per frequency step. While the Rayleigh response is given by the intensity contrast function of filters A and B according to Eq. (1), the Mie response is described by the centroid position of the Fizeau fringe according to Eq. (2). Polynomial fitting is then performed for each individual range gate to derive polynomial coefficients that are later fed into the wind retrieval algorithm (Marksteiner, 2013). Here, a fifth-order polynomial was empirically chosen for fitting the Rayleigh response curves, whereas a linear fit is applied for the Mie response function:

$$\Psi_{\text{Ray}}(f) = \sum_{i=0}^{5} c_i f^i, \tag{3a}$$

$$\Psi_{\text{Mie}}(f) = C_0 + C_1 f. \tag{3b}$$

The determined polynomial coefficients for each range gate are then used for the calculation of the Doppler frequency shift from the Rayleigh and Mie responses obtained for each wind observation. Since both the range gate setting and the flight altitude generally differ between the calibration flight and the actual wind scene, a linear interpolation is performed between the coefficients deduced from the calibration in order to obtain the response function for the respective bin altitudes of the wind observation. For the satellite instrument, the response function is derived for only one atmospheric range gate, thus necessitating a correction that considers the pressure and temperature profiles which affect the line shape of the molecular return signals (Dabas et al., 2008; Tan et al., 2008). Unlike for molecular scattering, the backscattering of the laser radiation from aerosols, cloud particles or hard targets does not induce a significant spectral broadening, so that the altitude-dependent variations in temperature- and pressure have a negligible impact on the Mie response. Therefore, in contrast to the Rayleigh response calibration, the Mie response function determined for the ground return is sufficient for the wind retrieval and used for all the atmospheric range gates. Due to this fact, precise determination of the coefficients $\{C_0, C_1\}$ for the ground is of utmost importance for an accurate Mie wind retrieval. A detailed study on A2D response calibrations and the various influencing factors that affect their quality is provided in Marksteiner et al. (2018). Based on a set of criteria which have been defined over the last years, out of the six available from 2016 one particular calibration, i.e. set of response coefficients $\{c_i\}$ ($i = 1, ..., 5$) and $\{C_0, C_1\}$, was determined as the baseline for the subsequent Rayleigh and Mie wind retrieval.

## 3.2 Refined ground detection scheme

Precise identification of the ground return signals is crucial for exploiting the information included therein. Systematic wind errors which can be caused by changes in the alignment of the transmit-receive path or inaccuracies in the aircraft attitude data, can be reduced by applying ZWC. Regarding the aircraft speed of the Falcon, the specification of the incorporated GPS receiver assures an accuracy of better than 0.1 m·s⁻¹ (Weissmann et al., 2005). Due to the coarse vertical resolution



(hundreds of metres) of the A2D and ALADIN, ZWC based on ground return signals is rather challenging, as the ground bin is very likely to be contaminated by atmospheric signals. For the Mie channel, strong aerosol backscatter close to the ground can influence the ground speed measurement, while the SNR of the ground measurement for the Rayleigh channel is diminished by the broad bandwidth molecular return collected from near the ground surface. Moreover, both channels are

potentially affected by surface winds which introduce systematic errors in the measurement of the ground speed or sea surface with non-zero ground speed (Li et al., 2010). This situation is aggravated by the fact that the ground signals can be distributed over multiple range bins. First, this is due to the charge transfer process of the ACCD which leads to a temporal overlap in the acquisition of two subsequent range gates of about 1 μs. Laser timing fluctuations in combination with charge transfer inefficiency during the readout of the ACCD, especially occurring at high signal intensities, can cause a signal

spread over even more than two range gates within a measurement and observation. Second, varying ground elevations during the duration of one measurement (0.4 s, 20 pulses at 50 Hz repetition rate) and laser pointing fluctuations can lead to the detection of ground signals in multiple range gates, taking into account that the laser pulses cover a distance of 80 m along track on the ground at an aircraft speed of 200 m·s⁻¹. Figure 3 illustrates this circumstance for two cases; one with ground signals completely contained in one range gate (a) and another with ground signals distributed over two range gates

(b). The height difference between a reference ground elevation during one measurement and the upper bin border of the highest (or first) range gate that contains ground signals is denoted by $\Delta H$ and represents a measure of the atmospheric contribution to the ground signal detected by the A2D. The reference ground elevation per measurement is derived from the digital elevation model (DEM) ACE2, providing elevation data at a resolution of 9 arc seconds (300 m x 300 m at the equator) (Berry et al., 2010).

In previous A2D studies, ground detection for the calibration mode was based on a visual inspection of intensities per observation where layers containing ground signal were specified *per flight leg*. The signal intensities in the identified range gates were then summed up (Marksteiner et al., 2013). This approach leads to an underestimation of the actual ground signal and to an additional summation of atmospheric signal causing error-prone ground data, especially for varying terrain during the flight leg. The imperfect differentiation between atmospheric and ground return signals thus introduces systematic errors

in the ground response functions of both detection channels. Concerning the Mie channel, this affects the entire wind profile, as the ground response is used for the wind retrieval in all atmospheric range gates as mentioned above. Consequently, the ground detection scheme was refined for being applied to the complex terrain scenes encountered especially during NAWDEX.

In order to derive more accurate ground speeds, a trade-off has to be found between summing up as much ground signal as

possible and minimising the atmospheric portion in the ground bins. For this purpose, a ground detection algorithm *on measurement level* was developed (Weiler, 2017). Similar to the wind retrieval algorithm employed for Aeolus (Reitebuch et al., 2017a; Reitebuch et al., 2017b), it is based on a signal-gradient approach to estimate ground bin candidates within a predefined range around the ground level which is given by the DEM. In a range of ±3 bins around the expected ground level according to the DEM, the signal gradients of two adjacent bins are calculated for each measurement and per range gate $i$:




$$\frac{\Delta I_i}{\Delta R_i} = \frac{I_{i+1} - I_i}{R_{i+1} - R_i}.$$ (4)

Here, $I$ denotes integrated signal intensity per measurement, while $R$ is the range from the instrument to the bin centre which can be calculated from the respective range gate integration time. In a next step, threshold gradients are introduced to identify the uppermost and lowermost ground bin. For the analysed flights, thresholds of $T_{GR,high} = 15$ and $T_{GR,low} = -15$ have

been empirically found to yield consistent results for both the Rayleigh and Mie channel. In order to avoid large atmospheric contribution to the ground signal, another threshold $T_{GR;DEM+1}$ has been implemented which analyses the signal level of the range gate just above the (DEM) bin covering the reference ground elevation. If the intensity in this bin does not make up more than five percent of the total summed ground signal, it is not considered for the ground signal summation. Careful analysis has shown that ground intensities falling below that threshold have negligible influence on the accuracy of ground

response calibration curves or ground wind speeds and thus can be omitted for the ground signal summation (Weiler, 2017). Using this approach, $\Delta H$ and hence the atmospheric portion of the ground signal can be significantly diminished. The ground detection method has been employed for the analysis of the Mie and Rayleigh response calibration data obtained in the NAWDEX campaign and formed the basis for the ZWC applied for the wind scenes on 4 October 2016 discussed in section 4.2. Moreover, the comparison between refined ground detection and the previous scheme allows for the characterisation of

the influence of the atmospheric contamination of the ground calibration parameters.

The largest influence of the refined scheme on the calibration parameters compared to the former approach was obtained for the sixth response calibration procedure performed during NAWDEX on 15 October 2016 between 17:24 UTC and 17:48 UTC. The Rayleigh and Mie signal intensities measured during the calibration are shown in Fig. 4(a) and (b), respectively. The calibration flight was carried out in the region around 65.5°N and 17.8°W, which is characterized by a

mountainous and ice-free terrain with ground elevations ranging from about 200 m to 1200 m. Consequently, ground signals were detected in four different range gates (#20 to #23) during the calibration procedure, as the Falcon aircraft flew circular patterns over this region. While the ground response calibration based on the old ground detection method would have summed up all the signals contained in these four range gates for each observation, i.e. frequency step of the calibration, the refined method only considers those bins per measurement that fulfil the threshold conditions as explained above. The

corresponding Rayleigh and Mie ground masks illustrating the range bins that were identified as ground bins for each measurement are depicted in Fig. 4(c) and (d). Due to the different sensitivities of the two receiver channels, and thus different measured signal intensities, the two masks are not fully identical.

For both channels, the atmospheric contribution is drastically reduced resulting in more accurate response values. While the mean value of $\Delta H$ over all measurements of calibration #6 is 454 m and 505 m for the Rayleigh and Mie channel when the

old ground detection technique is applied, it is only 207 m and 249 m for the new method, respectively. An overview of the atmospheric contributions (mean $\Delta H$) for all the six Rayleigh and Mie response calibrations (RRC and MRC) using the two different ground detection schemes is given in Tables 2 and 3. The tables also summarise the zero- and first-order polynomial coefficients $\{c_0, c_1\}$ and $\{C_0, C_1\}$ (referred to as intercept and slope) obtained from fitting of the response curves





according to Eq. (3a) and Eq. (3b). The second- and higher-order coefficients $\{c_i\}$ ($i$ = 2, 3, 4, 5) of the Rayleigh response function are not given. Since calibration #1 was carried out using a different setting of the co-alignment loop reference position ($CoG_X/CoG_Y$) (see section 2.1) affecting the incidence angle of the backscattered signals on the Rayleigh and Mie spectrometer, the resulting calibration parameters were disregarded in the statistical calculations leading to the values

provided in Tables 2 and 3.

In general, larger deviations in the slope and intercept values between the two methods are present for the Rayleigh channel. This can be explained by the fact that the broadband Rayleigh channel is more sensitive to the broadband atmospheric molecular background signal than the narrowband Mie channel where the broadband atmospheric contribution leads to a nearly constant intensity offset to the narrowband ground signals. The impact on the Rayleigh channel is especially large in

cases of low albedo surfaces where the atmospheric contribution to the weaker ground signals is more pronounced. As a result, large discrepancies between the calibration parameters obtained with the old and new method are observed for the two last calibrations that were performed over ice-free land with low albedo in the UV. In particular, the intercept values derived for the RRC #6 discussed before differ by as much as $1.24 \cdot 10^{-2}$. Using a typical Rayleigh response slope value of $4.6 \cdot 10^{-4}$ $MHz^{-1}$ (Table 2) and the conversion between Doppler frequency shift and LOS wind speed (1 m·s$^{-1}$ ≙ 5.63 MHz)

introduced in section 2.2, this difference in intercept translates to a wind speed difference of 4.8 m·s$^{-1}$. That means that ground speed values determined from RRC #6 using either the old or the new ground detection method would differ by that value. With a view to ZWC, the large discrepancy in the ground speed values underlines the relevance of proper ground detection for the wind retrieval, as the ground speeds are used as zero reference for the derived wind speeds. Likewise, using the refined ground detection method for the analysis of MRC #6 results in a change in the Mie intercept values by $11.7 \cdot 10^{-3}$

pixel which corresponds to a wind speed difference of 0.2 m·s$^{-1}$, considering a typical Mie response slope of about 100 MHz/pixel (Table 3).

Another aspect that becomes obvious from Tables 2 and 3 is that the spread of intercept values between the different Rayleigh response calibrations is reduced when applying the new ground detection method. The standard deviation over the five RRCs #2 to #6 is $1.02 \cdot 10^{-2}$, whereas it is $0.68 \cdot 10^{-2}$ for the new method. Hence, depending on the calibration used for the

25 wind retrieval, the Rayleigh ground wind speed varies by 3.9 m·s$^{-1}$ if the old technique is applied. This value is reduced by more than 30% to 2.6 m·s$^{-1}$ with the new scheme which is still unsatisfactorily large regarding the consistency of Rayleigh response calibrations. For the Mie channel, no change in the spread of the calibration parameters is evident. Nevertheless, the new ground detection approach provides a considerable improvement in the accuracy of the ground calibration parameters and, in turn, of the derived ground wind speeds. With a view to the Aeolus mission, it can be concluded that calibrations

should be performed over surfaces with high albedo, like ice surfaces, in order to minimise the impact of the atmospheric contamination. Furthermore, the quantity $\Delta H$ could be considered as a quality parameter for evaluating the quality of response calibrations or even to correct calibrations for the atmospheric contribution.



## 4 Wind retrieval and assessment of accuracy

This chapter discusses the wind results from two selected flights performed on 27 September and 4 October 2016 to demonstrate the Rayleigh and Mie wind retrieval algorithms as well as their subsequent validation by statistical comparison with the data obtained with DLR's coherent reference wind lidar system.

### 4.1 Jet stream wind observations over the North Atlantic on 27 September 2016

While the instrument response calibrations were performed during two dedicated flights over Iceland, the other 12 research flights within the NAWDEX campaign were devoted to wind observations over the North Atlantic region. Here, sampling of the jet stream was of particular interest with regards to both the pre-launch activities of Aeolus and the scientific objectives related to atmospheric dynamics. The observation of high horizontal wind speeds and large wind gradients occurring in relation to the jet provided an extensive characterization of the instrument over a large operating range and accurate wind profiles for the NAWDEX science objectives. In the context of the fourth NAWDEX intensive observation period, the overarching goal of the flight carried out on 27 September 2016 was to observe very high jet stream wind speeds related to the former tropical cyclone "Karl". As "Karl" moved towards the mid-latitudes, it merged with an initially weak downstream cyclone and strongly intensified. Later, at the time of the flight, the already weakened cyclone was located between Iceland and Scotland and the zonally oriented jet stream extended towards Scotland with horizontal wind speeds exceeding 80 m·s$^{-1}$ at altitudes of 9 to 10 km (see Fig. 5 and for a detailed description of the meteorological situation refer to Schäfler et al., 2018). To observe the high wind speeds, the Falcon aircraft flew towards the Faroe Islands and the Outer Hebrides right into the centre of the jet stream at a flight altitude of 11.5 km before returning to the air base in Keflavík. The satellite image taken from the Moderate Resolution Imaging Spectroradiometer (MODIS) instrument aboard NASA's Terra satellite (MODIS, 2017a), shown in Fig. 5(a), depicts increased cloud coverage along the flight track crossing the cyclone. From the total flight duration of three hours and 56 minutes (09:28 to 13:24 UTC), wind observations were conducted in the period between 10:28 and 12:36 UTC, split into two scenes of about one hour each.

### 4.1.1 Rayleigh background subtraction and quality control

In the period from 11:41 to 11:47 UTC the A2D was operated at a different mode which aimed at the detection of the Rayleigh background signal on the Mie channel. Proper quantification of the broadband molecular return signal transmitted through the Fizeau interferometer is important for avoiding systematic errors in the determination of the fringe centroid position and, in turn, in the Mie winds. Therefore, the laser frequency was tuned away by 1.8 GHz from the Rayleigh filter cross point which defines the nominal set frequency during the wind scenes (see Fig. 2(a)). In this way, the laser frequency of the emitted pulses was outside of the useful spectral range of the Mie spectrometer, so that the fringe was not imaged onto the Mie ACCD and only the broadband Rayleigh signal was detected on the Mie channel. The range-dependent intensity levels per pixel were subsequently subtracted from the measured raw Mie signal. In the near-field range gates, the measured



intensity distribution over the pixel array measured by the Mie and Rayleigh ACCDs is substantially impacted by the central obscuration of the telescope pupil by the secondary mirror and its supporting spider. Furthermore, the data obtained from the near-field region is affected by the incomplete overlap of the transmitted laser beam with the telescope field-of-view as well as by the attenuation of the signals by the EOM (Paffrath et al., 2009). Therefore, the atmospheric range gates in the region

within 1.5 km below the aircraft (range gates #5 and #6) were not considered in the wind retrieval.

The Rayleigh as well as the Mie signal intensities after Rayleigh background correction per observation (18 s) are shown in Fig. 6(a) and (b), respectively. The raw signals were first corrected for the DCO and the solar background which are collected in two separate range gates. Moreover, a range correction was applied taking into account that the intensity decreases as the inverse square of the distance between the scatterer and the detector. Finally, the integration times set for

each range gate were considered for normalising the signal intensities per bin. Curve flights during the flight section are manifested in altitude variations of the range gate borders, as a change in the roll angle of the aircraft involved a change in the off-nadir angle of the A2D. While the intensity profiles for the Rayleigh channel essentially follow the vertical distribution of the atmospheric molecular density, the Mie intensity profiles display the vertical distribution of atmospheric cloud and aerosol layers along the flight track. High Rayleigh signal intensities above 3.5 arbitrary units (a.u.) (dark red bins

in Fig. 6(a)) can be attributed to cloud layers at different altitudes along the flight track which also manifest in increased Mie signal intensities (Fig. 6(b)).

As a preparatory step of the wind retrieval, several quality control mechanisms were applied to exclude invalid data. The detection of corrupted measurements within one observation involved the screening for DCO outliers, saturated pixels on the ACCDs as well as for failure of the trigger that initiates the detector electronics. The latter causes an untimely ACCD

acquisition, and hence an incorrect allocation of the internal reference and atmospheric return signals to their designated range gates. For the actual wind retrieval, the wind speeds for each atmospheric range gate were determined from the respective frequency differences to the internal reference frequency. The frequencies were calculated from the corresponding Rayleigh and Mie response functions (Eqs. (3a) and (3b)) derived during the calibration mode. As a result, separate wind profiles for the Rayleigh and Mie channel were obtained. While the Rayleigh profiles only contain valid wind data in range

bins in which purely molecular backscattering occurred, the Mie wind profiles are composed of wind data retrieved from areas with sufficient cloud and aerosol content. However, since the retrieval initially produces wind values for all data bins in both channels, additional measures had to be taken to identify and eliminate invalid wind data. The procedures differ between the Rayleigh and Mie profiles and will be outlined in the following sections.

## 4.1.2 Rayleigh wind profiles

The identification of invalid winds retrieved from the Rayleigh channel was based on the detection of bins which were affected by particulate backscatter from clouds or aerosols, since this Mie contamination introduces systematic errors of the measured Rayleigh response (Dabas et al., 2008). Therefore, as introduced in Marksteiner (2013), bins showing range- and integration time-corrected Rayleigh signal intensities that are unusually high for pure molecular backscatter were excluded





from further analysis. An intensity threshold of 0.1 a.u. per measurement was found to be an appropriate value for identifying Mie-contaminated bins in the Rayleigh channel. Under clear conditions Rayleigh signal intensities on observation level (summed over 35 measurements) are well below 3.5 a.u. (see Fig. 6(a)). Due to the attenuation of the laser beam during propagation through the clouds, the wind information obtained from the range gates below clouds is very likely

to be also derogated. Consequently, not only the cloud bins themselves are flagged invalid but also all the bins in the range-gates below. Additionally, ground bins that were detected by the scheme described in section 3.2 as well as bins containing valid Mie wind data (see next section) were removed from the Rayleigh wind profiles.

Figure 7(a) shows the processed LOS Rayleigh winds plotted versus time and altitude for the period from 10:28 UTC to 12:36 UTC after removal of invalid bins as described above. During the first section of the flight, the horizontal component

of the A2D LOS unit vector was nearly parallel to the horizontal wind vector and pointing towards the wind, resulting in high positive LOS wind speeds (yellow/orange colours), whereas negative wind speeds of comparable magnitude were measured during the second flight leg when the LOS unit vector was oriented along the direction of the wind, i.e. the wind was pointing away from the instrument (blue colours). The data gap in between is due to the curve flight near the Outer Hebrides as well as the procedure required for Rayleigh background subtraction mentioned above. The figure also illustrates

the range-dependent vertical resolution of the instrument. For the presented flight section, the integration time of the ACCD was set to 2.1 µs in the range gates #8 to #14 (9.4 km to 7.7 km) and those close to the ground (#22, #23); 4.2 µs in the range gates #7, #15 and #16 (6.1 km); and 8.4 µs in all the remaining atmospheric range rates, corresponding to a height resolution of 296 m, 592 m and 1184 m, respectively. This range gate setting was the same for the Rayleigh and Mie channel and chosen in order to resolve the wind structure within the core of the jet stream. In this region, broad coverage of Rayleigh

winds was obtained, while mid-level clouds prevented the acquisition of valid Rayleigh wind data on the edges of the jet below their tops between 4 km and 7 km height. In addition, high-level clouds at the beginning of the shown flight section limited the extension of the Rayleigh wind profiles to the range from 9 km to 10 km.

One characteristic of the Rayleigh channel is the fluctuating wind error from profile to profile, which becomes visible as a vertical texture in the two-dimensional wind curtain. The underlying reason is the high sensitivity of the Rayleigh response

to variations in the incidence angle on the FPI. Despite the active transmit-receive co-alignment loop, residual angular variations in the order of a few µrad, which are due to atmospheric turbulence and the effect of strong cloud backscatter onto the co-alignment algorithm, cause fluctuations in the derived wind speeds of several $m \cdot s^{-1}$. The introduced error is identical for all the atmospheric range gates, and the mean error varies from observation to observation, resulting in a vertical pattern in the Rayleigh wind curtain. Measures are being examined to reduce this fluctuation by a refined co-alignment feedback

loop, for instance, by employing a UV camera with higher resolution in combination with an improved algorithm for determining the centre of gravity of the backscattered laser radiation.



### 4.1.3 Mie wind profiles

The validity of the Mie wind determined for each bin is related to the cloud and aerosol loading in the respective range gate, and thus the signal intensity detected on the Mie ACCD. For the proper identification of bins with sufficient particulate backscatter return signal, the so-called Mie SNR was defined as the quotient between the signal of the pixel with the highest

intensity, i.e. the fringe centre, and the mean over the pixels that lie outside the fringe (Marksteiner, 2013). The Mie SNR calculated for the studied measurement scene is depicted in Fig. 6(c). Based on the SNR profile, a threshold value was set which allowed sorting out corrupt wind measurement bins. For the analysed wind scene, a Mie SNR threshold of 5.0 was empirically chosen in order to remove those bins where low particle backscatter coefficients prevented the correct determination of the Mie fringe centroid position and thus the acquisition of accurate wind speeds.

The resulting two-dimensional Mie wind curtain is shown in Fig. 7(b). As opposed to the Rayleigh channel, the Mie data coverage is rather sparse owing to the little cloud cover and low aerosol load during the flight. Wind data is mainly obtained from the cloudy regions mentioned above, thus complementing the wind information gained with the Rayleigh channel. The combination of the Rayleigh and Mie wind data, displayed in a composite curtain in Fig. 7(c), illustrates the complementarity of the two detection channels which enables the acquisition of wind speeds under various atmospheric

conditions, hence ensuring broad data coverage for the entire scene. In the case that valid winds are obtained for both channels, the Mie wind is preferred due to the generally lower systematic and random error (see next sections). Figure 8(a) shows the combined Rayleigh and Mie wind curtain along two flight legs in the region of the jet stream. Here, the horizontal LOS (HLOS) wind speed is illustrated which was calculated from the measured LOS wind speeds and the off-nadir angle of the instrument ($\approx20°$) per observation. Strong vertical wind gradients exceeding 10 m·s$^{-1}$/km at about 5 km altitude become

apparent in Fig. 8(b) depicting the HLOS wind profiles from two selected observations which started at 11:28:21 UTC and 11:54:09 UTC, respectively. The vertical position of the data points corresponds to the altitude at the centre of the respective range bin. HLOS wind speeds above 80 m·s$^{-1}$ were measured in the centre of the sampled jet stream, which is in agreement with the modelled wind field shown in Fig. 5, considering the difference in the angle between the HLOS unit vector of the A2D and the horizontal wind vector.

### 4.1.4 Coherent wind lidar as reference system

Validation of the A2D instrument performance and wind retrieval algorithms was performed by comparing the resulting wind profiles to those obtained with DLR's well-established coherent wind lidar system emitting at 2 µm wavelength and 500 Hz repetition rate, which was operating in parallel on board the Falcon aircraft providing accuracy of the horizontal wind speed of better than 0.1 m·s$^{-1}$ and precision of better than 1 m·s$^{-1}$ (Weissmann et al., 2005; Chouza et al., 2016b). In

contrast to the A2D, the determination of the Doppler shift by the 2-µm lidar system relies on heterodyne detection using the instruments' seed laser as local oscillator (Witschas et al., 2017) and thus does not rely on any calibration procedures. Moreover, the coherent wind lidar incorporates a scanner which allows retrieving the three-dimensional horizontal wind




vector from a number of LOS wind measurements with a vertical resolution of 100 m. For this purpose, the instrument performs conical scans at an off-nadir angle of 20°, while the information from 21 azimuthal positions is used for the wind vector retrieval. On each azimuthal position the signal from 500 laser pulses (1 s) is averaged to obtain one LOS profile. The time for positioning the laser at its scan starting position is around 21 s resulting in a total time of 42 s for one observation of the 2-µm wind lidar, whereas one A2D observation takes 18 s as outlined above.

For adequate comparison of the wind profiles measured with the 2-µm and the A2D wind lidar, the three-dimensional wind vectors had to be projected onto the A2D LOS axis. This was carried out for each 2-µm observation by calculating the scalar product of the measured wind vector and the mean A2D LOS unit vector under consideration of the aircraft attitude during the respective observation period. Furthermore, the different spatial and temporal resolutions of the two wind lidar instruments necessitated an adaptation of the 2-µm measurement grid to that of the A2D. This was accomplished by a weighted aerial interpolation algorithm (Marksteiner et al., 2011). Here, one considers the whole two-dimensional A2D wind curtain overlaid by the 2-µm grid. Hence, a single A2D bin can be covered by multiple 2-µm bins both horizontally and vertically. The overlapping regions form a new composite 2-µm bin. The contributions of the single 2-µm winds to the wind value allocated to the composite bin are weighted by the overlap of the respective 2-µm bins with the regarded A2D bin. In this way, the A2D and 2-µm wind profiles can be compared on a bin-to-bin basis.

In order to reduce the risk of large discrepancies between the interpolated 2-µm wind and the compared A2D wind in case of low coverage, a minimum overlap of the compared bins (coverage ratio threshold) has been introduced as a quality control parameter. For the considered wind scene, a threshold value of 25% was found to provide an optimal trade-off between comparability and quantity of the 2-µm bins, thus yielding an acceptable number (nearly 1000) of representative composite 2-µm bins used for comparison. Increasing the coverage ratio threshold, e.g. to 80%, would have reduced the number of bins to less than 500 without significant change in the parameters resulting from the statistical comparison. Furthermore, proper analysis of the Rayleigh winds with a sufficient number of compared bins (>300) required a threshold of less than 45%.

The projected LOS wind curtain obtained from the 2-µm DWL after adaptation to the A2D measurement grid is depicted in Fig. 7(d). Since the 2-µm DWL purely relies on particulate backscatter, the data coverage is similar to that of the A2D Mie channel, resulting in a large overlap of the two data types. Consequently, the number of bins available for comparison is greater than for the Rayleigh channel. However, the availability of 2-µm wind data from the upper region of the jet stream between 9 km and 10 km altitude allows for the comparison of Rayleigh wind data over a broad range of wind speeds.

### 4.1.5 Statistical comparison of A2D and 2-µm DWL winds

The statistical comparison of the Rayleigh and Mie winds with the 2-µm DWL data from the discussed flight section is visualised in Fig. 9(a). Here, the A2D winds are plotted versus the corresponding interpolated 2-µm winds, resulting in a cloud of data points that ideally lie on the dashed line representing $v_{A2D} = v_{2\mu m}$. The non-weighted linear fit $v_{A2D} = A \cdot v_{2\mu m} + B$ through the real data provides values for the slope $A$ and intercept $B$ that generally deviate from the ideal result $A = 1$ and $B = 0$. Six bins with wind speed differences $v_{A2D} - v_{2\mu m}$ larger than ±10 m·s$^{-1}$ were identified as gross errors





in the Rayleigh dataset and thus removed from the sample. The statistical values derived from the scatterplot are summarised in Table 4, showing that the fitting parameters for both Rayleigh and Mie only slightly deviate from the ideal case ($A \approx 1$, $|B| < 0.5$ m·s$^{-1}$). The standard error of the slope given in the table was calculated according to

$$s_A = \sqrt{\frac{\frac{1}{n-2}\sum_{i=1}^{n} \varepsilon_i^2}{\sum_{i=1}^{n}(v_{2\mu m,i} - \overline{v_{2\mu m}})^2}}, \text{ with} \tag{5a}$$

$$\varepsilon_i = v_{A2D,i} - (A \cdot v_{2\mu m,i} + B) \tag{5b}$$

being the residuals of the linear regression. It should be noted that the parameters derived from the statistical comparison are influenced by the systematic and random errors of both the A2D and the 2-µm lidar. However, since the latter provides high accuracy and precision as stated above, the total errors are dominated by the systematic and random error of the A2D.

Aside from the standard deviation, the median absolute deviation (MAD) was determined as an additional parameter for
evaluating the random error of the A2D wind speed measurements. It is defined as the median of the absolute variations of the measured wind speeds from the median of the wind speed differences:

$$\text{MAD} = \text{median}\left[\left|(v_{A2D,i} - v_{2\mu m,i}) - \text{median}(v_{A2D,i} - v_{2\mu m,i})\right|\right]. \tag{6}$$

The MAD represents a robust measure of the variability of the measured wind speeds and is more immune to outliers compared to the standard deviation σ. If the random wind error is normally distributed, the MAD value is related to the
standard deviation as σ ≈ 1.4826 · MAD. The latter quantity is referred to as *scaled MAD*.

The scatterplot illustrates the good agreement of the A2D and 2-µm DWL data over the range of LOS wind speeds from -22 m·s$^{-1}$ to +26 m·s$^{-1}$. For both detection channels the correlation coefficient is as high as $r = 0.97$. Aside from the different wind speed span, the Rayleigh and Mie winds primarily differ with respect to the mean bias ($v_{A2D} - v_{2\mu m}$) over all data points representing the accuracy of the instrument. Here, the Mie wind bias almost vanishes (-0.03 m·s$^{-1}$), which is due
to the fact that the A2D winds are nearly symmetrically distributed about the reference 2-µm winds, leading to positive and negative deviations of similar magnitude which compensate each other.

For the Rayleigh winds, a negative bias of -0.49 m·s$^{-1}$ is obtained, resulting in a mean bias of the combined Rayleigh and Mie data of about -0.21 m·s$^{-1}$. The corresponding HLOS wind speed bias of -0.61 m·s$^{-1}$ (= -0.21/sin(20°)) is considered to be adequate with regards to the Aeolus mission where absolute HLOS mean bias values better than 0.7 m·s$^{-1}$ are required.
However, it should be noted that the mean bias shows larger values when considered per range gate, as depicted in Fig. 8(b). The extreme bias values >3 m·s$^{-1}$ in range gates #8 to #10 lack statistical significance, as they result from a very small number of compared bins due to the scarce data coverage of the 2-µm DWL at altitudes between 8.5 km and 9.5 km. For the other range gates, the mean bias varies between -0.7 m·s$^{-1}$ and 0.3 m·s$^{-1}$.

Another important statistical parameter for the evaluation of the instrument performance is the standard deviation which
represents the random error, and hence the precision of the A2D. Here, the Mie winds show a value of 1.5 m·s$^{-1}$ (HLOS: 4.3 m·s$^{-1}$) which is beyond the requirements of Aeolus. In order to meet the mission goals, the satellite instrument should provide a precision of 1 m·s$^{-1}$ in the planetary boundary layer, 2.5 m·s$^{-1}$ in the troposphere and 3 m·s$^{-1}$ to 5 m·s$^{-1}$ in the




stratosphere (ESA, 2016). The random error can also be approximated from probability density functions (PDFs) illustrating the frequency distribution of the wind speed differences $v_{A2D} - v_{2\mu m}$, i.e. the wind error, for the Rayleigh and Mie channel (see Fig. 9(b) and (c)). For the Mie channel, the wind random error is nearly Gaussian-distributed, while a number of outliers with $v_{A2D} - v_{2\mu m} \approx 6$ m·s$^{-1}$ leads to a discrepancy between the mean bias (-0.03 m·s$^{-1}$) and the centre of the Gaussian fit (-0.08 m·s$^{-1}$). For the same reason, the e$^{-1/2}$-width of the fit ($2w = 2.7$ m·s$^{-1}$) is narrower than twice the standard deviation ($2\sigma = 3.0$ m·s$^{-1}$) which also considers the outliers. Finally, due to the deviation from a Gaussian distribution, the scaled MAD of 1.3 m·s$^{-1}$ is slightly smaller than $\sigma$.

Speckle noise was identified as one of the major causes for the increased random error of the A2D Rayleigh and Mie channel. The noise is introduced by the use of a fibre to transmit the internal reference signal from the laser to the front optics where it is injected into the receiver reception path and co-aligned with the atmospheric signal, as shown in Fig. 1. This is different compared to the free optical path set-up in the transceiver of the satellite instrument which does not suffer this difficulty. The speckle pattern which was estimated to consist of about only 2000 speckles is the input for the Fizeau spectrometer and, after modification by reflection, also for the Fabry-Pérot spectrometers (DLR, 2016). Although the speckle pattern is static over short time scales of a few seconds to minutes, slow changes in the intensity distribution of the internal reference signal are introduced by variations in laser frequency, polarization or (ambient) fibre temperature, which in turn modify the response of the Mie and Rayleigh spectrometers. Since the response measured for the internal reference forms the basis for the determination of the Doppler frequency shift, and thus, the wind speed in each atmospheric range gate, the speckle-induced fluctuations increase the random error over the entire wind profile. Comparisons of the internal reference frequencies derived from the Rayleigh and Mie responses against the frequencies measured using the wavemeter showed random variations ($2\sigma$) in the order of 8 MHz (Mie) and 11 MHz (Rayleigh), corresponding to LOS wind errors of 1.4 m·s$^{-1}$ and 2.0 m·s$^{-1}$, respectively. Effective speckle reduction is envisaged, e.g. by incorporating a moving diffuser into the beam path of the internal reference signal in order to rapidly change the speckle pattern within one observation, thus averaging out the variations. Another contribution to the random error in the A2D Mie channel is the occurrence of a heterogeneous cloud structure. In particular, the position of the top edges of optically thick clouds within one range gate has a significant influence on the wind data. According to Sun et al. (2014) who investigated the performance of Aeolus in heterogeneous atmospheric conditions using high-resolution radiosonde data, a non-uniform distribution of clouds and/or aerosols within a range bin introduces random errors in the Mie HLOS winds of several m·s$^{-1}$ depending on the bin size and altitude.

Besides the speckle noise and the impact of the atmosphere, a further contribution to the random error of the Mie winds is caused by an imperfect response calibration procedure using a linear fitting function to describe the relationship between the Doppler frequency shift and the position of the fringe produced by the Fizeau interferometer. Hence, a more adequate fitting function will be applied in the future in order to take into account the Mie response nonlinearities and to improve the precision of the Mie channel.

For the Rayleigh channel the random error is even larger ($\sigma = 2.7$ m·s$^{-1}$). Apart from the speckle noise in the internal reference signal, the error contributions are different than for the Mie channel. The Rayleigh response calibration considers





nonlinearities by using a fifth-order polynomial function for fitting the response curve. However, the measurement principle based on the double-edge technique using a sequential FPI is much more sensitive to angular variations of the backscattered light compared to the fringe-imaging technique employed in the Mie channel. As explained above, small angular fluctuations of 1 µrad with respect to the 200 mm-diameter telescope with a FOV of 100 µrad introduce variations in the measured LOS

wind speeds of about 0.4 m·s$^{-1}$ (DLR, 2016). Furthermore, the availability of 2-µm wind data in those range bins that were used for the evaluation of the Rayleigh winds suggests at least a small contamination of the Rayleigh signal by particulate backscatter, thus introducing an increased random error (Dabas et al., 2008). Like for the Mie channel, the PDF for the Rayleigh wind random error exhibits slight deviations from a Gaussian distribution. Consequently, the value 1.4826 · MAD = 2.6 m·s$^{-1}$ marginally differs from σ = 2.7 m·s$^{-1}$.

**4.2 Zero wind correction for the flight on 4 October 2016**

The wind scene on 27 September 2016 presented in the previous sections was characterized by optically dense clouds at different altitudes. As a consequence, the ground return signals detected during the scene were too weak for reliable determination of the ground speed which could be used for ZWC. Consequently, for this particular research flight, the refined ground detection scheme could not be exploited for reducing the systematic error of the Mie and Rayleigh wind

speeds. Unfortunately, this circumstance holds true for most of the flights conducted in the context of NAWDEX, since the flight planning was primarily driven by the atmospheric science objectives of the campaign, resulting in complex atmospheric conditions with rather dense cloud coverage. One exception is the flight performed on 4 October 2016, which was dedicated to the investigation of the jet stream east of Iceland. For this purpose, the Falcon aircraft crossed the jet stream with increased wind speeds twice, as it flew two legs back and forth between the way points located at 66.0'N, 17.5°W and

64.0°N, 7.0°W (see Fig. 10). To the west of the jet axis cloud-free conditions prevailed over the northeast of Iceland. Hence, high ground visibility was obtained at the beginning of the first leg and at the end of the second leg, as can be seen by the visible satellite image (MODIS, 2017b) a few hours after the flight depicted in Fig. 10(a) together with the flight track of the Falcon. The A2D measured wind profiles during the periods from 09:00 to 09:44 UTC and from 09:54 to 10:30 UTC (see also Table 1). The figure reveals the contrasting atmospheric circumstances experienced during the flight which were

characterized by highly variable cloud cover along the flight path.
Using the same Rayleigh and Mie response calibrations as for the flight on 27/09/2016, the results of the wind retrieval are displayed in Fig. 11. While the Rayleigh wind curtain shows good coverage at the beginning and the end of the period (Fig. 11(a)), valid Mie winds were primarily obtained in the vicinity of the jet stream centre which was sampled in the middle of the flight (Fig. 11(b)). This again underlines the complementarity of the two channels which allows for excellent

data coverage despite strongly diverse atmospheric conditions. Since the direction of the wind was towards the A2D LOS on the first leg, positive LOS wind speeds of up to 25 m·s$^{-1}$ (HLOS: 73 m·s$^{-1}$) were measured, whereas negative winds of the same magnitude were detected on the flight leg back to Iceland.





The systematic and random errors for the Rayleigh and Mie winds were determined from a statistical comparison with the 2-µm reference wind lidar data. The resulting scatterplots and PDFs are shown in Fig. 12, while the statistical parameters are given in Table 5. Due to the poor overlap of the A2D Rayleigh wind data with the 2-µm wind curtain (see Fig. 11(d)), only a small number of data points (168) entered the comparison despite a low coverage ratio threshold of 25%. Consequently, the

calculated mean bias (1.54 m·s$^{-1}$) and scaled MAD (2.7 m·s$^{-1}$) lack statistical significance. This becomes also obvious from the shape of the histogram illustrating the distribution of the Rayleigh wind errors (Fig. 12(b)) which strongly deviates from a Gaussian distribution. For this reason, the following discussion concentrates on the Mie channel. Here, a scaled MAD of 2.0 m·s$^{-1}$ was derived from the comparison with the reference lidar which showed large data overlap with the Mie channel, resulting in 1246 compared bins. The mean bias of 0.57 m·s$^{-1}$ is considerably larger than the value obtained for the flight on

27/09/2016. The increase in systematic error might result from changes in the alignment of the transmit-receive path which slightly varies from flight to flight. In combination with potential inaccuracies in the aircraft attitude data, this leads to unknown contributions to the retrieved LOS wind speed which are not considered in the retrieval algorithm.

The wind speed offset can, however, be reduced by ZWC based on the developed ground detection scheme. Any deviation from zero is interpreted as systematic error in the wind speed retrieval and hence subtracted from the measured wind speed.

The ground speed (or ZWC) values obtained for the Mie channel during the two wind scenes on 04/10/2016 are plotted in Fig. 11(c). From a total number of 268 observations, 59 observations included valid ZWC values in the ground range gates which were identified by the algorithm explained in section 3.2. The respective observations are indicated as grey boxes in the Mie wind curtain. Thanks to the refined ground detection on measurement level, atmospheric contamination of the ground signals was minimized, thus ensuring that the detrimental influence of near-surface winds on the ZWC values was

diminished. The mean of the ZWC values was determined to be 0.53 m·s$^{-1}$ with a standard deviation of 1.2 m·s$^{-1}$. The variation around the mean which is also observed as random error in the atmospheric Mie wind speeds can again be traced back to fluctuations in the Mie response measured for the internal reference. In order to confirm the correlation between the variability of the ZWC values and the internal reference variations, the Mie responses of the internal reference were converted to relative (laser) frequencies using the Mie response calibration. The obtained frequencies were compared to the

frequencies measured with the high-precision wavemeter which tracked the absolute wavelength of the laser pulses emitted during the flight. The frequency difference (Mie response minus wavemeter) was finally translated into wind speed differences (1 m·s$^{-1} \cong 5.63$ MHz), resulting in the dashed line plotted in Fig. 11(c). The course of the curve is obviously correlated to the progression of the ZWC values, thus verifying that the noise in the internal reference considerably affects the measured ground speeds. As mentioned in the previous section, speckle noise is responsible for Mie response variations

in the order of $\sigma = 0.7$ m·s$^{-1}$. Nevertheless, the mean value was used for correcting the Mie wind speeds, leading to the scatterplot depicted in Fig. 12(a). The statistical parameters after ZWC are given in the right column Table 5. Subtraction of the mean ZWC value reduces the mean bias to 0.04 m·s$^{-1}$ which is comparable to the result obtained for the flight on 27/09/2016. Hence, ZWC in combination with the refined ground detection scheme improves the accuracy of the A2D remarkably for the discussed flight.



## 5 Summary and Conclusion

The ALADIN Airborne Demonstrator (A2D) represents an essential testbed for the validation of the upcoming Aeolus mission. Due to its similar and representative design and operation principle, the A2D provides valuable information on the wind measurement strategies of the satellite instrument as well as on the optimization of the wind retrieval and related quality-control algorithms. For this purpose, the A2D was successfully deployed for wind observations in the international airborne field campaign NAWDEX conducted in Iceland in autumn 2016. Within the scope of the campaign, 14 research flights were performed extending the wind and calibration dataset of the A2D for validating the retrieval algorithms and operation procedures. In particular, the recording of very high HLOS wind speeds above 80 m·s$^{-1}$ was obtained by sampling the North Atlantic jet stream, while the complementarity of the Rayleigh and Mie channel allowed for broad vertical and horizontal coverage across the troposphere.

Comparison of the A2D wind data with a high-resolution coherent Doppler wind lidar emitting at 2 µm wavelength enabled the evaluation of the performance of the A2D in terms of accuracy and precision. For the flight on 27/09/2016, the mean bias was found to be -0.49 m·s$^{-1}$ for the Rayleigh channel and -0.03 m·s$^{-1}$ for the Mie channel. A larger Mie wind speed bias of 0.57 m·s$^{-1}$ was determined for the flight on 04/10/2016, but could be reduced to 0.04 m·s$^{-1}$ by means of ZWC. The latter was supported by accurate ground detection using a scheme that minimizes the contribution of atmospheric return signals in the identified ground range gates. This method was also implemented in the analysis of the Rayleigh and Mie response calibrations where it is particularly effective in case of low-albedo surfaces in the UV (e.g. land) or areas with strongly varying ground elevations. The ground detection scheme is envisaged to be fully exploited in upcoming airborne campaigns to provide accurate ZWC for flights with sufficient ground visibility. In order to reduce the random error both in the detected ground speeds and in the atmospheric wind speeds, the response fluctuations in the internal reference signals need to be diminished. This problem, which is absent in the satellite instrument, is proposed to be solved by avoiding slow variations in the speckle pattern incident on the Mie and Rayleigh spectrometers, e.g. by implementing a fast diffuser.

In addition to the internal reference fluctuations, the large random errors of about 2.7 m·s$^{-1}$ in the Rayleigh channel can be traced back to the transmit-receive path co-alignment in combination with the high incidence angle sensitivity of the Rayleigh spectrometer, while the heterogeneity of the atmosphere and the nonlinearity of the Mie response function are considered to be additional factors contributing to the random error (1.5 m·s$^{-1}$) observed for the Mie winds. Hence, apart from the technical development of the A2D regarding speckle reduction and improved co-alignment, the main focus of the current research is on the improvement of the system accuracy and precision by implementing a novel Mie response calibration procedure considering nonlinearities. The modifications of the A2D are intended to be tested in the frame of forthcoming airborne campaigns which will also aim at conducting flights in coordination with the Aeolus satellite after its launch in 2018.





## Acknowledgements

The development of the ALADIN Airborne Demonstrator and the work carried out during the NAWDEX campaign were supported by the German Aerospace Center (Deutsches Zentrum für Luft- und Raumfahrt e.V., DLR) and the European Space Agency (ESA), providing funds related to the preparation of Aeolus (WindVal II, contract No. 4000114053/15/NL/FF/gp), as well as NRL Monterrey and the EUropean Facility for Airborne Research (EUFAR, project NAWDEX Influence). The authors are especially grateful to Engelbert Nagel for his constant support throughout the campaign.

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



# Figures

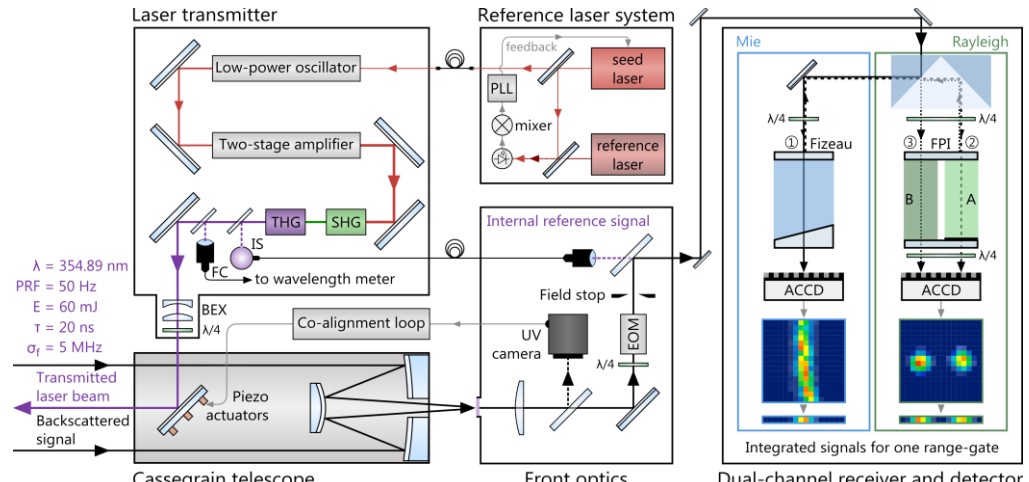

**Figure 1.** Schematic of the ALADIN Airborne Demonstrator (A2D) wind lidar instrument consisting of an injection-seeded, frequency-tripled laser transmitter, a Cassegrain telescope, front optics and a dual-channel receiver. PLL: phase locked loop, SHG: second harmonic generator, THG: third harmonic generator, IS: integrating sphere, FC: fiber coupler, BEX: beam expander, EOM: electro-optic modulator, FPI: Fabry-Pérot interferometer, ACCD: accumulation charge-coupled device.

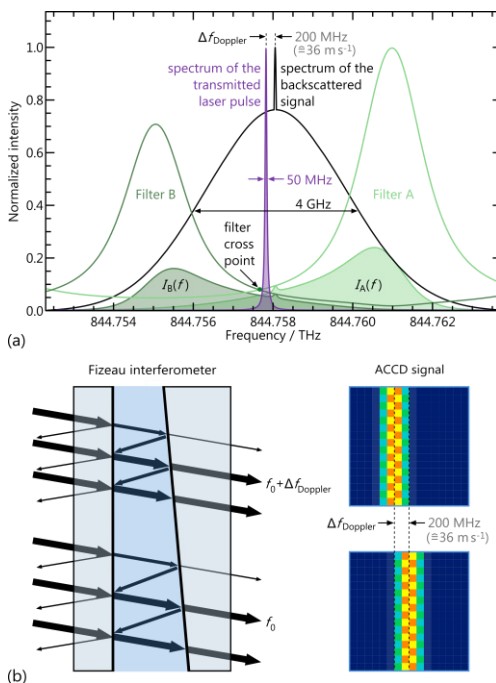

**Figure 2.** (a) Spectral distribution of the transmitted laser pulse (purple) and the backscattered signal (black) which is composed of the narrowband Mie and the broadband Rayleigh component. The transmission spectra of the two FPI filters of the Rayleigh channel are shown in green, while the filled areas illustrate the respective transmitted intensities $I_A(f)$ and $I_B(f)$ for determining the Doppler shift. (b) Operation principle of the Mie channel based on the fringe-imaging technique.





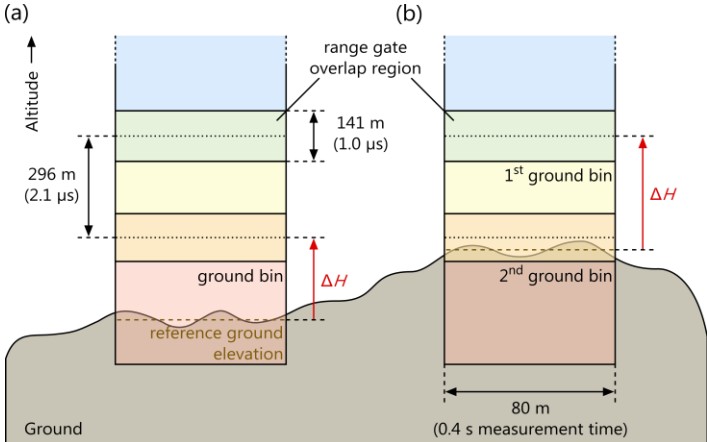

**Figure 3.** Detection of ground signals with the A2D wind lidar. The sketch shows the vertical position of three neighbouring range gates (blue, yellow and red boxes) with respect to the ground. The ground return signals are either contained in only one range bin (a) or distributed over two range bins due to the range gate overlap (here shown for the Mie channel as green and orange areas) as well as varying elevation of the ground surface within one measurement (b). $\Delta H$ denotes the atmospheric contribution to the signal obtained from the ground bin(s). The given heights of 296 m and 141 m are related to the A2D off-nadir angle of 20°.

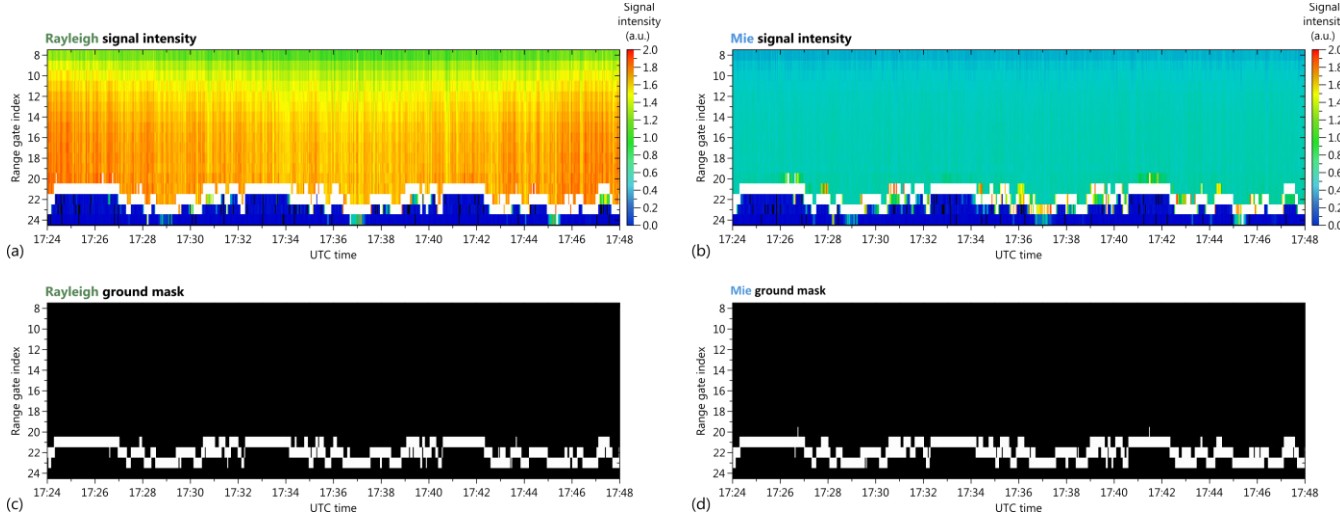

**Figure 4.** Ground detection during the response calibration performed over Iceland on 15 October 2016 between 17:24 and 17:48 UTC. (a) Signal intensities measured with the A2D Rayleigh channel versus time and the range gates #8 to #24. (b) Mie signal intensity including Rayleigh background. The intensities are range-corrected and scaled to the integration time of the respective range rates. Range gates #8 to #19 have a length of 592 m, while range gates #20 to #24 have a length of 296 m. Bins with signal intensities exceeding the maximum of the respective colour scale are printed in white. The Rayleigh and Mie ground masks resulting from the developed ground detection scheme are depicted in (c) and (d), respectively. Orange bins are identified as ground bins and thus considered for the determination of the ground response function.





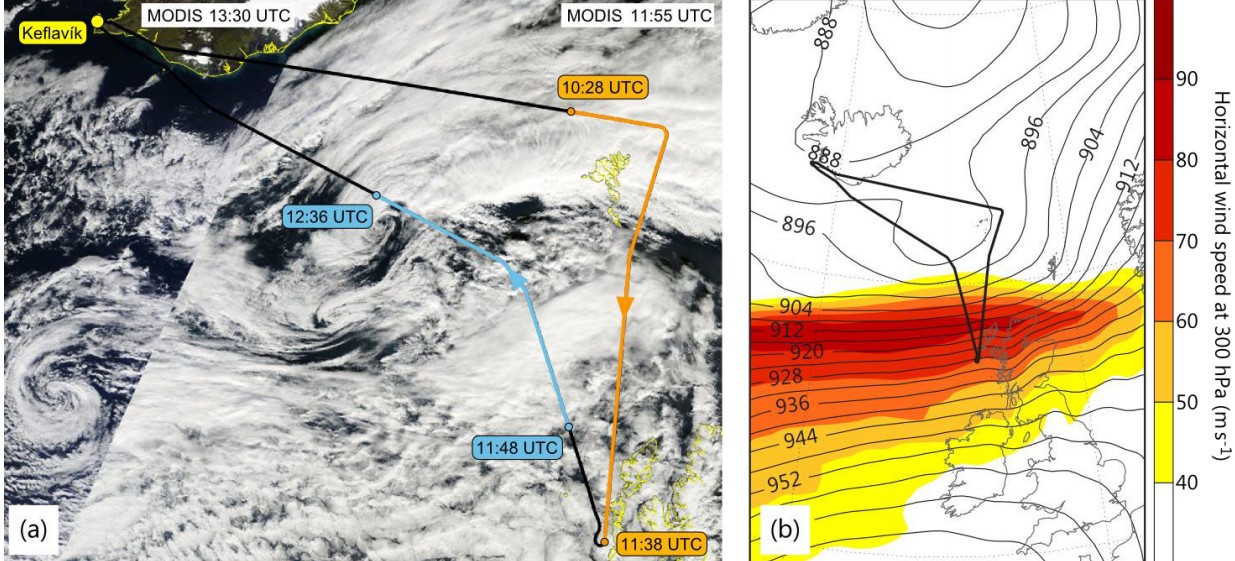

**Figure 5.** (a) Flight track of the Falcon aircraft (black line) during the research flight conducted on 27 September 2016. The wind scenes performed from 10:28 UTC to 11:38 UTC and from 11:48 UTC to 12:36 UTC are indicated in orange and blue. The background picture is composed of a map provided by Google Earth and satellite images from Terra MODIS (VIS channel) taken at 11:55 UTC (right part) and 13:30 UTC (left part) (MODIS, 2017a). (b) Geopotential Height (black isolines, m) and horizontal wind speed (colour shading) at 300 hPa on 27 September 2016, 12 UTC from ECMWF model analysis together with the flight track of the Falcon 20 aircraft.




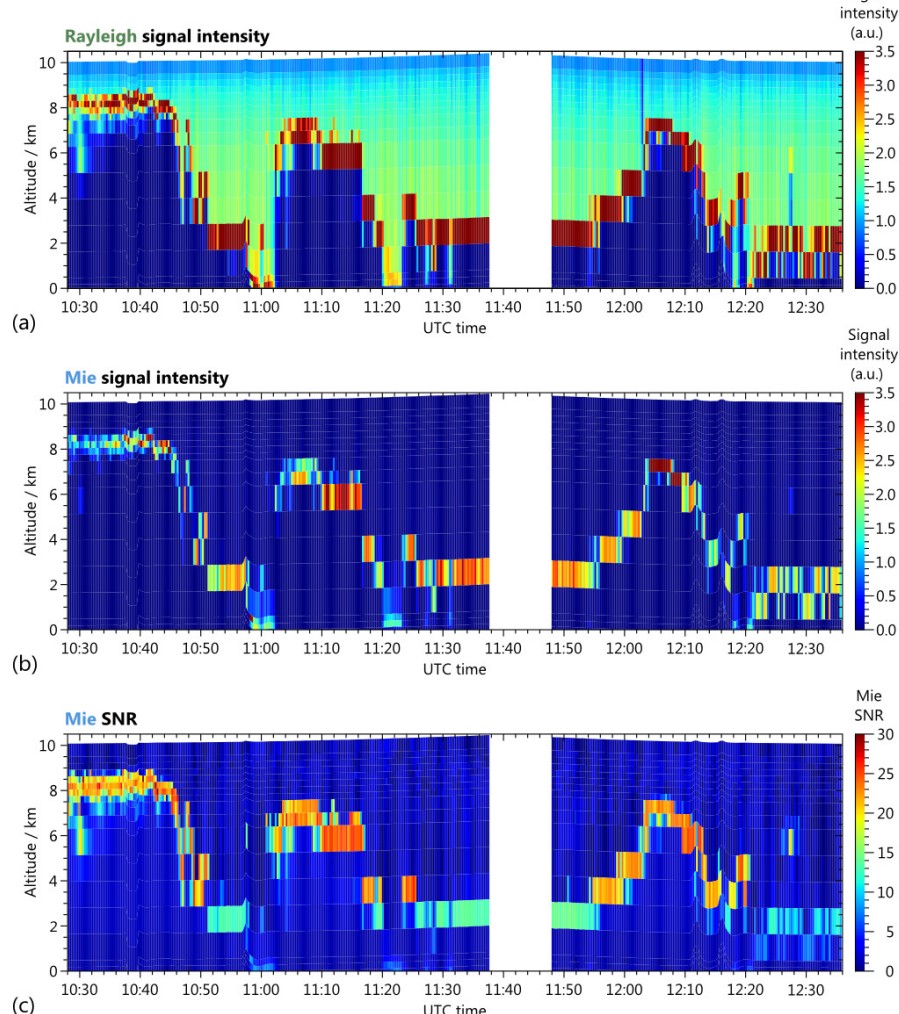

**Figure 6.** Signal intensities measured for (a) the A2D Rayleigh channel and (b) the A2D Mie channel during the flight on 27 September 2016 between 10:28 UTC and 12:36 UTC. The intensities are range-corrected and scaled to the integration time of the respective range-rates. The background and detection chain offset were subtracted. For the Mie channel, the Rayleigh background signal was subtracted as explained in the text. The detection of the Rayleigh background signal was performed between 11:41 UTC and 11:47 UTC leading to a data gap in this period. (c) Mie SNR calculated according to Eq. (3.29) in Marksteiner (2013). Bins with signal intensities exceeding the maximum of the respective colour scale are printed in dark red.





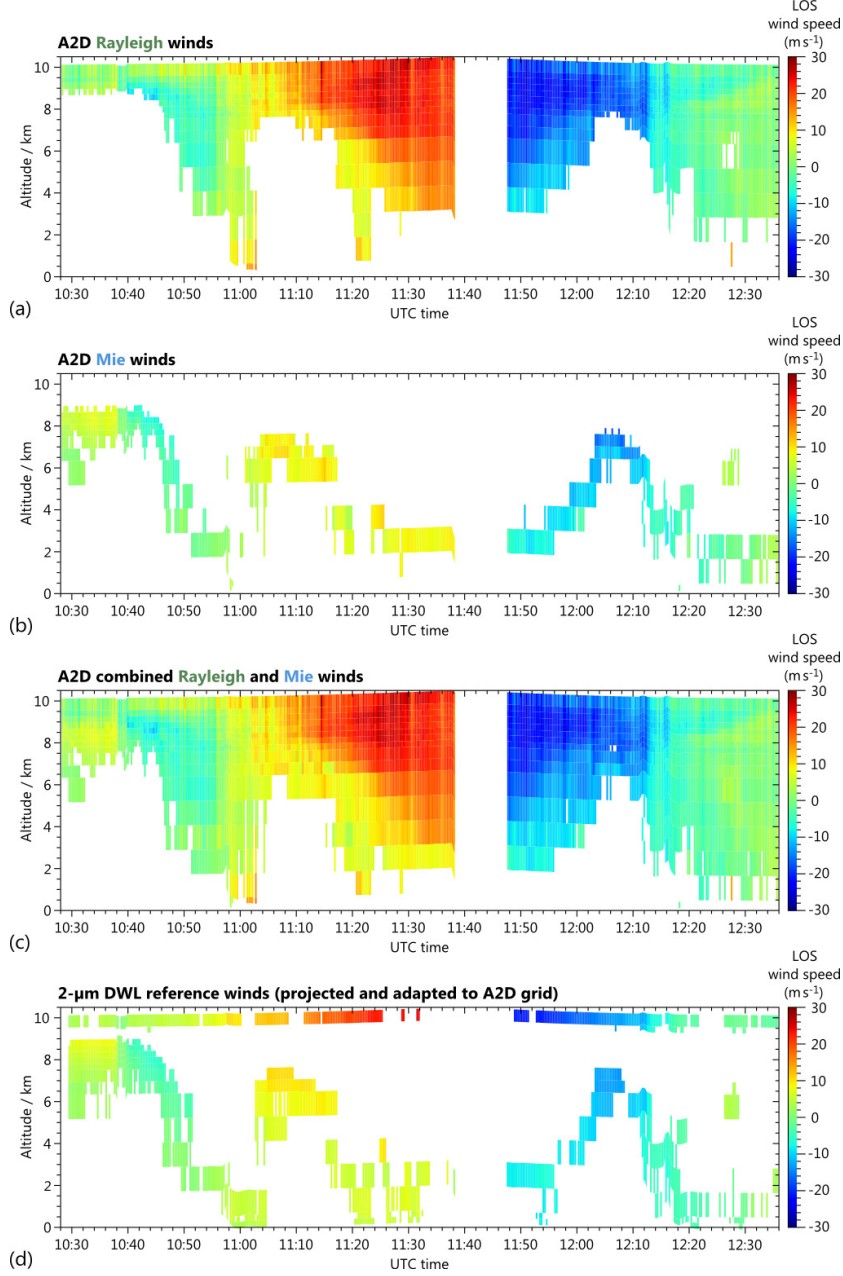

**Figure 7.** LOS wind profiles (positive towards the instrument) measured during the flight on 27 September 2016 between 10:28 UTC and 12:36 UTC using (a) the A2D Rayleigh channel and (b) the A2D Mie channel. The combination of both channels is depicted in (c), while (d) shows the corresponding wind curtain obtained with the coherent 2-µm reference wind lidar. For better comparison, the 2-µm wind data were adapted to the measurement grid of the A2D. White colour represents missing or invalid data due to low signal, e.g. in case of low aerosol loads or below dense clouds. The data gap between 11:38 UTC and 11:48 UTC is due to an interruption of the wind measurement during a curve flight and a different operation mode of the A2D instrument aiming at the detection of the Rayleigh background signals on the Mie channel.





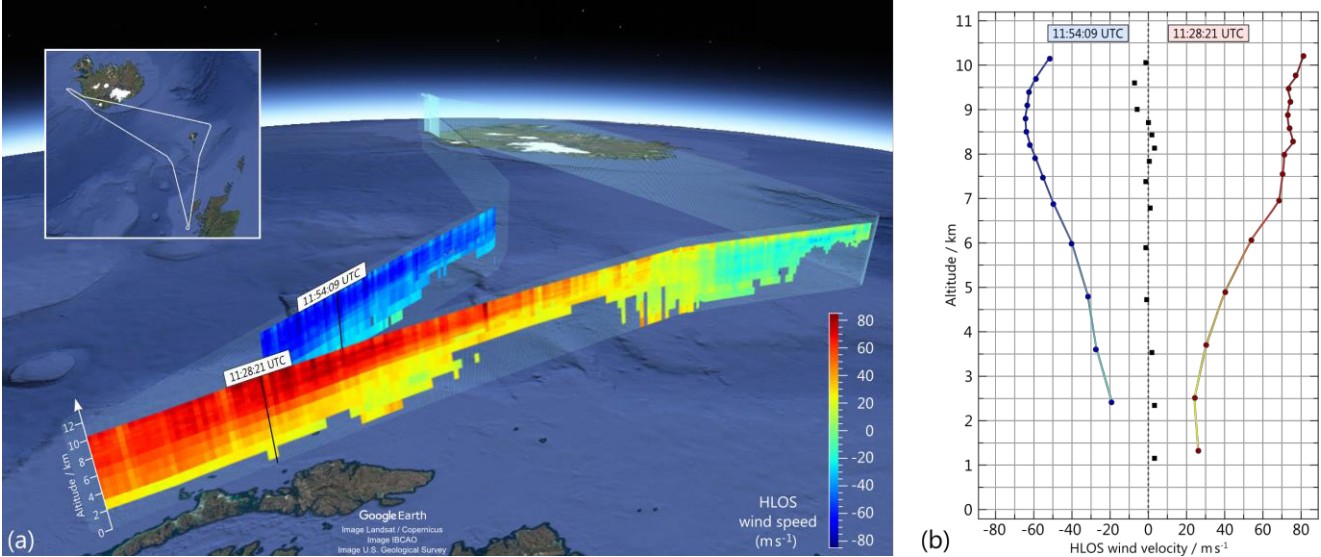

**Figure 8.** Flight track of the Falcon 20 aircraft for the research flight on 27 September 2016 together with the overlaid A2D HLOS wind profiles measured between 10:40 UTC and 11:38 UTC (foreground) as well as between 11:48 UTC and 12:12 UTC (background), whilst crossing the North Atlantic jet stream (background image: © 2017 Google). (b) Wind profiles from two selected observations starting at 11:28:21 UTC and 11:54:09 UTC. The black squares indicate the mean bias per range gate based on the comparison with wind data from the 2-µm coherent wind lidar (see text).

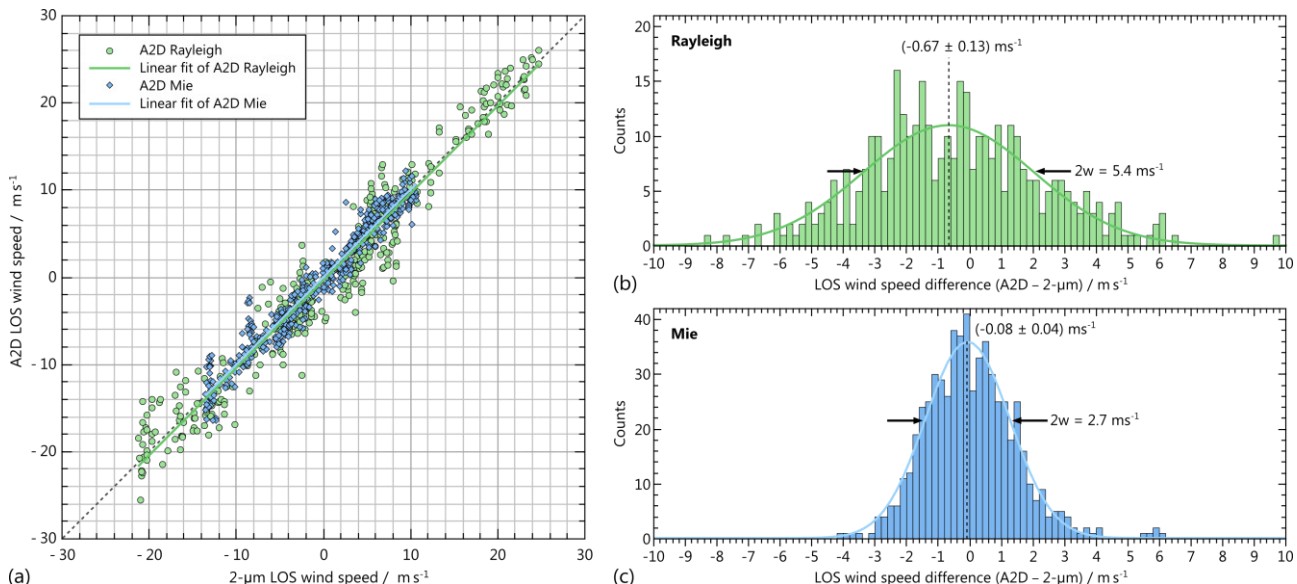

**Figure 9.** (a) A2D LOS wind speed determined with the Rayleigh (dots) and Mie (diamonds) channel versus the 2-µm LOS wind speed for comparison of the wind data measured during the flight on 27 September 2016 between 10:28 UTC and 12:36 UTC (see corresponding curtains in Fig. 7(a), (b) and (d). The scatterplot is obtained by adaptation of the different measurement grids of the two systems based on a weighted interpolation algorithm and a subsequent bin-to-bin comparison. The corresponding probability density functions for the wind differences (A2D – 2-µm) are shown in (b) and (c) for the Rayleigh and Mie channel, respectively. The solid lines represent Gaussian fits with the given centres and $e^{-1/2}$-widths $2w$.




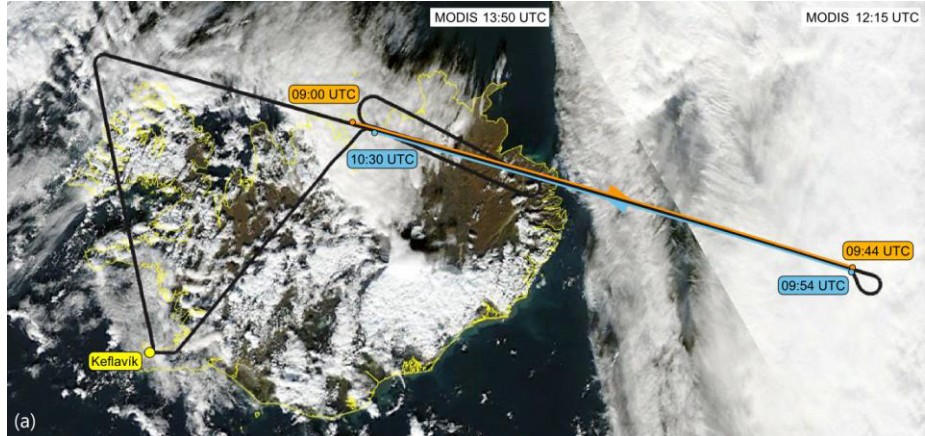
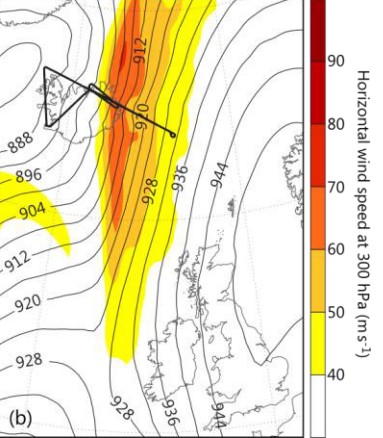

**Figure 10.** (a) Flight track of the Falcon aircraft (black line) during the research flight conducted on 4 October 2016. The wind scenes performed from 09:00 UTC to 09:44 UTC and from 09:54 UTC to 10:30 UTC are indicated in orange and blue. High ground visibility was obtained over the northeast of Iceland at the beginning and the end of the scenes, respectively. The background picture is composed of a map provided by Google Earth and satellite images from Aqua MODIS (VIS channel) taken at 12:15 UTC (right part) and 13:50 UTC (left part) (MODIS, 2017b). (b) Geopotential Height (black isolines, m) and horizontal wind speed (colour shading) at 300 hPa over the North Atlantic on 27 September 2016, 12 UTC from ECMWF model analysis together with the flight track of the Falcon 20 aircraft.





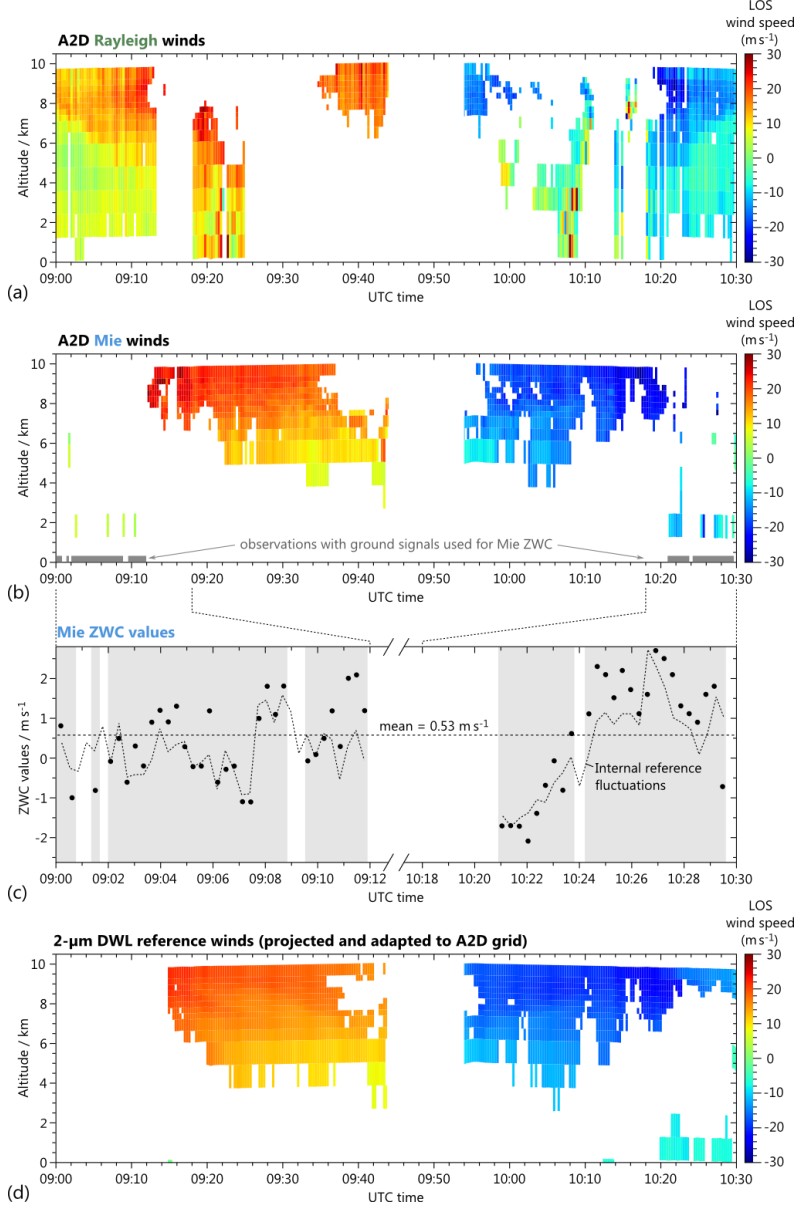

**Figure 11.** LOS wind profiles (positive towards the instrument) measured during the flight on 4 October 2016 between 09:00 UTC and 10:30 UTC using (a) the A2D Rayleigh channel and (b) the A2D Mie channel. The grey boxes indicate periods during which the ground visibility was sufficient for obtaining ZWC data. The corresponding ZWC values are plotted in (c) together with the ground speed variations introduced by the Mie response fluctuations in the internal reference signals (see text). (d) Wind curtain measured with the coherent 2-μm reference wind lidar. For better comparison, the 2-μm wind data were adapted to the measurement grid of the A2D. The data gap between 09:44 UTC and 09:54 UTC is due to an interruption of the wind measurement during a curve flight.





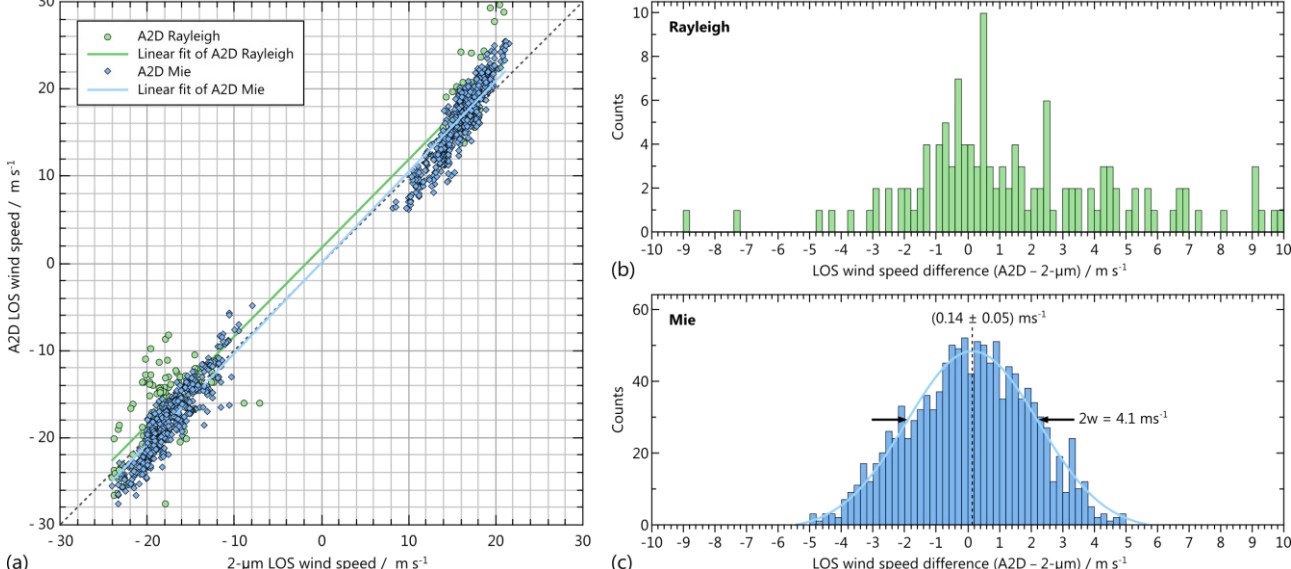

**Figure 12.** (a) A2D LOS wind speed determined with the Rayleigh (dots) and Mie (diamonds) channel versus the 2-µm LOS wind speed for comparison of the wind data measured during the flight on 04 October 2016 between 09:00 UTC and 10:30 UTC (see corresponding curtains in Fig. 11(a), (b) and (d). The scatterplot for the Mie channel was obtained after Zero Wind Correction was applied to the measured wind speeds. The corresponding probability density functions for the wind differences (A2D – 2-µm) are shown in (b) and (c) for the Rayleigh and Mie channel, respectively. The solid lines represent Gaussian fits with the given centres and $e^{-1/2}$-widths $2w$.




## Tables

**Table 1.** Overview of the research flights of the Falcon aircraft conducted in the frame of the NAWDEX campaign and the wind scenes performed with the A2D. The flights on 27/09/2016 and 04/10/2016 discussed in the present work are printed in bold. The two flights on 28/09/2016 and 15/10/2016 were dedicated to response calibrations of the Rayleigh and Mie channel (see section 3.1), while the first two and the last two flights on 17/09/2016 and 18/10/2016 were transfer flights between Oberpfaffenhofen, Germany and the airbase in Keflavík, Iceland.

| Flight # | Date | Flight period (UTC) | Measurement period (UTC) | Number of observations |
|---|---|---|---|---|
| 1 | 17/09/2016 | 06:10 – 08:07 | 06:59 – 07:12 | 44 |
|  |  |  | 07:34 – 07:45 | 38 |
| 2 | 17/09/2016 | 10:01 – 13:33 | 10:30 – 11:31 | 203 |
|  |  |  | 11:42 – 12:24 | 140 |
|  |  |  | 12:43 – 13:07 | 82 |
| 3 | 2109/2016 | 14:00 – 17:17 | 14:56 – 15:27 | 100 |
|  |  |  | 15:34 – 15:57 | 78 |
|  |  |  | 16:11 – 16:51 | 134 |
| 4 | 23/09/2016 | 07:01 – 10:21 | 07:51 – 08:53 | 206 |
|  |  |  | 09:14 – 09:53 | 130 |
| **5** | **27/09/2016** | **09:28 – 13:24** | **10:28 – 11:38** | **234** |
|  |  |  | **11:48 – 12:36** | **160** |
| 6 | 28/09/2016 | 10:56 – 14:19 | Calibration flight | |
| 7 | 02/10/2016 | 08:31 – 12:01 | 09:42 – 09:53 | 38 |
|  |  |  | 10:07 – 10:47 | 136 |
|  |  |  | 11:06 – 11:30 | 80 |
| **8** | **04/10/2016** | **08:09 – 11:43** | **09:00 – 09:44** | **147** |
|  |  |  | **09:54 – 10:30** | **121** |
|  |  |  | 10:35 – 10:49 | 48 |
| 9 | 04/10/2016 | 13:04 – 15:49 | 13:58 – 14:51 | 179 |
|  |  |  | 15:02 – 15:14 | 41 |
| 10 | 09/10/2016 | 15:44 – 19:24 | 16:41 – 17:15 | 113 |
|  |  |  | 17:24 – 17:54 | 99 |
|  |  |  | 18:18 – 18:58 | 138 |
| 11 | 15/10/2016 | 10:05 – 13:34 | 10:53 – 11:07 | 50 |
| 12 | 15/10/2016 | 15:24 – 18:44 | Calibration flight | |
| 13 | 18/10/2016 | 08:36 – 11:14 | 09:20 – 09:57 | 123 |
|  |  |  | 10:24 – 10:37 | 45 |
| 14 | 18/10/2016 | 12:39 – 14:30 | 13:33 – 13:53 | 67 |



**Table 2.** Rayleigh response calibration parameters obtained from the six calibrations performed on 28/09/2016 and on 15/10/2016. The zero- and first-order fitting parameters $c_0$ and $c_1$ were derived involving the old ground and new ground (GR) detection method (see text). The atmospheric contribution $\Delta H$ (see Fig. 3) has been averaged over the respective calibration period. Calibration #1 was performed using a different alignment of the lidar system and is thus excluded from the statistical calculations.

| RRC # | Date | Surface | Mean $\Delta H$ (m) | | Slope ($c_1$) ($10^{-4}$ MHz$^{-1}$) | | Intercept ($c_0$) ($10^{-2}$) | |
|---|---|---|---|---|---|---|---|---|
| | | | Old GR detection | New GR detection | Old GR detection | New GR detection | Old GR detection | New GR detection |
| 1 | 28/09/2016 | Ice | 480 | 308 | 4.58 | 4.43 | 1.15 | 1.39 |
| 2 | 28/09/2016 | Ice | 753 | 519 | 4.46 | 4.42 | 1.62 | 1.73 |
| 3 | 28/09/2016 | Ice | 734 | 522 | 4.48 | 4.44 | 1.60 | 1.70 |
| 4 | 15/10/2016 | Ice | 606 | 546 | 4.64 | 4.63 | 0.32 | 0.36 |
| 5 | 15/10/2016 | Ice-free land | 411 | 249 | 4.92 | 4.78 | -0.32 | 0.42 |
| 6 | 15/10/2016 | Ice-free land | 454 | 207 | 4.82 | 4.69 | -0.47 | 0.77 |
| Mean | | | 592 | 409 | 4.66 | 4.59 | 0.55 | 1.00 |
| Standard deviation | | | 157 | 166 | 0.20 | 0.16 | 1.02 | 0.68 |

**Table 3.** Mie response calibration parameters obtained from the six calibrations performed on 28/09/2016 and on 15/10/2016. The zero- and first-order fitting parameters $C_0$ and $C_1$ were derived involving the old ground and new ground (GR) detection method (see text). The atmospheric contribution $\Delta H$ (see Fig. 3) has been averaged over the respective calibration period. Calibration #1 was performed using a different alignment of the lidar system and is thus excluded from the statistical calculations.

| MRC # | Date | Surface | Mean $\Delta H$ (m) | | Slope ($C_1$) (MHz/pixel) | | Intercept ($C_0$) ($10^{-3}$ pixel) | |
|---|---|---|---|---|---|---|---|---|
| | | | Old GR detection | New GR detection | Old GR detection | New GR detection | Old GR detection | New GR detection |
| 1 | 28/09/2016 | Ice | 436 | 383 | -98.1 | -98.5 | -119 | -116 |
| 2 | 28/09/2016 | Ice | 729 | 613 | -97.9 | -97.9 | -116 | -116 |
| 3 | 28/09/2016 | Ice | 714 | 648 | -97.8 | -97.8 | -110 | -110 |
| 4 | 15/10/2016 | Ice | 601 | 570 | -98.0 | -97.9 | -44.9 | -44.3 |
| 5 | 15/10/2016 | Ice-free land | 384 | 274 | -96.6 | -96.6 | -70.9 | -69.6 |
| 6 | 15/10/2016 | Ice-free land | 505 | 249 | -98.0 | -97.9 | -82.1 | -93.8 |
| Mean | | | 562 | 456 | -97.7 | -97.8 | -90.5 | -91.6 |
| Standard deviation | | | 144 | 177 | 0.56 | 0.63 | 29.6 | 29.2 |





**Table 4.** Results of the statistical comparison between the A2D and the 2-µm LOS wind data measured on 27/09/2016. The statistical comparison has been performed for the Rayleigh and Mie wind profiles (see corresponding scatterplots in Fig. 9) as well as for the combined wind curtain as shown in Fig. 7(c).

| Statistical parameter | Rayleigh winds | Mie winds | Combined winds |
|---|---|---|---|
| Number of compared bins | 381 | 562 | 943 |
| Number of removed bins due to gross error (> ±10 m·s⁻¹) | 6 | 0 | 6 |
| Correlation coefficient $r$ | 0.97 | 0.98 | 0.97 |
| Slope $A$ | 1.002 ± 0.012 | 1.004 ± 0.009 | 1.002 ± 0.008 |
| Intercept $B$ | -0.49 m·s⁻¹ | -0.03 m·s⁻¹ | -0.21 m·s⁻¹ |
| Mean bias | -0.49 m·s⁻¹ | -0.03 m·s⁻¹ | -0.21 m·s⁻¹ |
| Standard deviation | 2.7 m·s⁻¹ | 1.5 m·s⁻¹ | 2.0 m·s⁻¹ |
| 1.4826 · median absolute deviation | 2.6 m·s⁻¹ | 1.3 m·s⁻¹ | 1.8 m·s⁻¹ |

5  **Table 5.** Results of the statistical comparison between the A2D and the 2-µm LOS wind data measured on 04/10/2016. The statistical comparison for the Mie wind profiles was performed without and with ZWC.

| Statistical parameter | Rayleigh winds | Mie winds (without ZWC) | Mie winds (with ZWC) |
|---|---|---|---|
| Number of compared bins | 168 | 1246 | 1246 |
| Number of removed bins due to gross error (>±10 m·s⁻¹) | 11 | 0 | 0 |
| Correlation coefficient $r$ | 0.96 | 0.99 | 0.99 |
| Slope $A$ | 1.01 ± 0.02 | 1.04 ± 0.03 | 1.04 ± 0.03 |
| Intercept $B$ | 1.67 m·s⁻¹ | 0.55 m·s⁻¹ | 0.02 m·s⁻¹ |
| Mean bias | 1.54 m·s⁻¹ | 0.57 m·s⁻¹ | 0.04 m·s⁻¹ |
| Standard deviation | 3.3 m·s⁻¹ | 1.9 m·s⁻¹ | 1.9 m·s⁻¹ |
| 1.4826 · median absolute deviation | 2.7 m·s⁻¹ | 2.0 m·s⁻¹ | 2.0 m·s⁻¹ |