# Peer review of "Airborne wind lidar observations over the North Atlantic in 2016 for the pre-launch validation of the satellite mission Aeolus"

_Atmospheric Measurement Techniques, 2018_

## Referee Comment (RC1) · G.-J. Marseille (Referee) · 1 Feb 2018

Review of paper "Airborne wind lidar observations over the North Atlantic in 2016 for the pre-launch validation of the satellite mission Aeolus" by Lux et. al.

The authors discuss aircraft campaigns with the A2D in preparation for launch of the first Doppler wind lidar in space: Aeolus. The processing of observed data with Aeolus to winds is known to be challenging in particular in dynamically complex scenes, including strong wind shear and varying cloud conditions. The measurements done in this campaign near Iceland are therefore of particular interest to the algorithm development teams to prepare for processing challenging real data from space to winds. In addition,

the importance of, and challenges for, zero wind calibration were demonstrated and an improved scheme discussed.

Overall, the paper is very well written and very clear. Also the differences between the A2D and the satellite (Aeolus) have been well explained.

General comments • The authors spend a substantial part of the paper on ground detection and zero wind calibration. Clearly, high quality calibration is crucial for the quality of the final product, the wind profile. Despite the proposed solution, calibration will still be challenging, also for a space borne instrument. Systematic errors due to imperfect differentiation between atmospheric and ground return signals will be hard to avoid. Is it true that the wind difference between adjacent bins (i.e. wind-shear) does not suffer from these systematic errors? In that case, rather than producing a wind profile, one could produce a profile of wind-shear for use in NWP, clearly at the expense of losing one bin, but without systematic errors from calibration issues. Can the authors please elaborate on this wind-shear option. • Wind-shear profiling may also resolve the curtain issue discussed on page 14: "The introduced error is identical for all the atmospheric range gates" • In section 4.1.4 the authors discuss the issue of comparing A2D Rayleigh winds and 2 micron lidar data. The fact that the 2 micron lidar does provide measurements between 9 and 10 km altitude suggests the presence of particles in this region and hence contamination of A2D Rayleigh winds. This explains part of the poorer statistics of A2D Rayleigh winds, as the authors correctly mention in section 4.1.5. Also, from Figure 9d, it appears that the range of wind speeds is largest in this area and thus largest wind variability. It may therefore be the most challenging region for wind measurements, where Mie winds have a relatively "easy job" further down in the troposphere. Considering, in addition, the height assignment error (unknown location and distribution of cloud and/or aerosols inside the bin) apparent for Mie winds in particular, the remark on page 15: "Mie wind is preferred due to the generally lower systematic and random error (see next sections)" may be too strong based on the presented results. Can the authors please comment on this? Also how these

conclusions translate to Aeolus? Can you please comment?

minor comments • Page 2, replace "as it will close the gaps in the wind data coverage" by "as it will contribute to close the gap in wind profile data coverage" • Page 2, line 19 "aircraft" => aircrafts • Page 5, line 14; replace "from moving particles (cloud particles, aerosols, molecules)" by "from particles (cloud droplets, aerosols) and molecules with move with the ambient wind" • Page 6, x0, x and k have not been clearly defined near equation 2. Please do. • Caption of figure 2: "the respective transmitted intensities the respective transmitted intensities". Should "transmitted" here not be replaced by "received"? • Caption of Figure 4. I do not understand the last sentence: "Orange bins are identified as ground bins and thus considered for the determination of the ground response function." Which orange bins? Please explain or correct.
* * *

---

## Referee Comment (RC2) · M. Hardesty (Referee) · 12 Apr 2018

This is a very well-written and thorough paper describing a flight campaign to validate techniques and instrumentation that will be used in the Aeolus mission to measure winds from space. The paper is important because it provides an indication of performance and potential issues associated with Aeolus measurements and also provides a direct comparison of winds measured by coherent and direct detection wind systems observing the same volume of atmosphere.

The only section of the paper for which I have some question was the part on use of the ground returns for correcting the velocity, which I found a bit unclear. My points on

this section are included in the list of comments shown below.

Specific comments on the manuscript, which I consider minor points, are as follows:

Page 3, line 30: Given that other characteristics of the transmitter are stated, it would be useful to know the bandwidth or frequency uncertainty of the transmitted pulse.

Page 4, line 3: Why are the receiver electronics triggered 60 microseconds before laser pulse emission?

Page 4, line 25: It wasn't clear to me exactly what the EOM does. Does it simply switch between the internal reference and the atmospheric signal after the pulse has been transmitted?

Page 8, line 18: It would be useful to know why the response function for the satellite instrument is only performed for a single range gate. How is that gate determined? Is the satellite attitude varied so that instrument looks vertically?

Page 9, line 20: I found this paragraph a bit confusing. What does "a visual inspection of intensities" mean, and why does a summation lead to an underestimation of the actual ground signal? Because the paper places quite a bit of emphasis on the improvement of the new technique for dealing with ground return, I think a better characterization of the old technique is necessary.

Page 10, line 31: I assume that delta h is computed from the DEM data for Table 2, although it isn't totally clear to me from the text.

Page 16, line 25: Because the coherent Doppler lidar measures return from aerosols, and the Rayleigh channel is sensitive to the presence of aerosols, is a comparison between the two valid? Doesn't this potentially overestimate the error in the Rayleigh winds, unless the effect of aerosol on that channel is negligible. Perhaps this should be discussed some more.

Page 17, line 1: The removal of the "bad" measurement from the Rayleigh dataset is

done based on comparison with the coherent lidar. How would this be handled for the satellite measurement, where no comparison source will be available, to avoid sending bad data to the assimilation algorithms?

Page 18, line: 23: Why exactly does a heterogeneous cloud structure contribute to random error? Is it a range-weighting effect in the presence of shear, or some other optical effect? Probably a bit more explanation is needed here.

Figure 4: The caption refers to "orange bins" in (c) and (d). I'm not sure what bins this refers to; I don't see any orange bins.

---

## Author Response (AR1)

***Responses to referee comments and changes to the manuscript***

*Airborne wind lidar observations over the North Atlantic in 2016 for the*
*pre-launch validation of the satellite mission Aeolus (https://doi.org/10.5194/amt-2018-19)*

***Response to Referee Comment 1***

Comment #1.1:

*The authors spend a substantial part of the paper on ground detection and zero wind calibration. Clearly, high quality calibration is crucial for the quality of the final product, the wind profile. Despite the proposed solution, calibration will still be challenging, also for a space borne instrument. Systematic errors due to imperfect differentiation between atmospheric and ground return signals will be hard to avoid. Is it true that the wind difference between adjacent bins (i.e. wind-shear) does not suffer from these systematic errors? In that case, rather than producing a wind profile, one could produce a profile of wind-shear for use in NWP, clearly at the expense of losing one bin, but without systematic errors from calibration issues. Can the authors please elaborate on this wind-shear option. Wind-shear profiling may also resolve the curtain issue discussed on page 14.*

Response to Comment #1.1:

Concerning systematic wind errors a distinction has to be made between range-independent and range-dependent error sources. First, systematic errors are caused by inaccuracies in the aircraft attitude angles, e.g. by improper knowledge of the laser pointing, or by constant errors in the wind retrieval, e.g. introduced by uncertainties in the calibration parameters. The resulting wind bias is constant along the wind profile and can be reduced by applying Zero Wind Correction, provided that sufficient ground return signals are available and that the atmospheric contamination of the ground return signals is low. If the latter conditions are not fulfilled, producing a wind-shear profile at the expense of one range bin is certainly an option for eliminating this systematic error source in the analysis of the airborne observations. Similar systematic error sources, e.g. improper knowledge of pointing direction or satellite-induced LOS speed, exist for the satellite instrument producing a slowly varying bias along the orbit which will be not present in wind-shear profiles. Such errors can be compensated by means of ZWC.

The second source of systematic wind errors which is specific to the airborne demonstrator is the imperfect transmit-receive co-alignment, as discussed on page 14. In contrast to the statement on page 14, the systematic error which varies from observation to observation and which manifests as vertical pattern in the Rayleigh wind curtain, is **not identical** for all the atmospheric range gates. It

is **only correlated** among all the atmospheric range gates, but also range-dependent. The error is largest in the near-field and decreases with increasing distance from the instrument, i.e. towards the ground. Thus, the curtain issue discussed on page 14 would not be fully resolved by plotting the wind-shear profile instead of the wind profile.

For the satellite instrument, the situation is more complicated due to the much higher ground track velocity of about 7.2 km/s. The different travel times of laser pulses backscattered from different altitudes in combination with the angular movement of the satellite during the propagation period of the pulses leads to range-dependent incidence angles of the backscattered light on the Rayleigh (and Mie) spectrometers, and hence to a range-dependent bias in the wind speeds. This effect will be characterized at the beginning of the Aeolus mission and can be subsequently corrected. Consequently, wind-shear profiles will not eliminate these range-dependent bias sources, if they are not fully corrected. In summary, it depends on the cause of the systematic error, if it can be eliminated with wind-shear profiles.

Changes to the manuscript #1.1:

The sentence in section 4.1.2 "The introduced error is identical for all the atmospheric range gates, and the mean error varies from observation to observation […]" was changed to "The introduced error is thus correlated among the atmospheric range gates, and the mean error varies from observation to observation […]". Moreover, a new section 4.1.6 was added discussing Rayleigh and Mie wind errors for the A2D and ALADIN. The section addresses the difference between range-dependent and range-independent errors as well as the wind shear option.

Comment #1.2:

*In section 4.1.4 the authors discuss the issue of comparing A2D Rayleigh winds and 2 micron lidar data. The fact that the 2 micron lidar does provide measurements between 9 and 10 km altitude suggests the presence of particles in this region and hence contamination of A2D Rayleigh winds. This explains part of the poorer statistics of A2D Rayleigh winds, as the authors correctly mention in section 4.1.5. Also, from Figure 9d, it appears that the range of wind speeds is largest in this area and thus largest wind variability. It may therefore be the most challenging region for wind measurements, where Mie winds have a relatively "easy job" further down in the troposphere. Considering, in addition, the height assignment error (unknown location and distribution of cloud and/or aerosols inside the bin) apparent for Mie winds in particular, the remark on page 15: "Mie wind is preferred due to the generally lower systematic and random error (see next sections)" may be too strong based on the presented results. Can the authors please comment on this? Also how these conclusions translate to Aeolus? Can you please comment?*

Response to Comment #1.2:

Regarding the comparison of Rayleigh winds with data from the 2-µm DWL, it should be mentioned that the latter is very sensitive even to weak aerosol backscatter return, due to its coherent detection principle with small bandwidth. In addition, the deployment of a coherent DWL on an aircraft is favourable, because the atmospheric altitudes with low aerosol backscatter are located in near range gates, which do not suffer from the $R^2$-dependency of the signal and strong aerosol extinction (as it would be the case for ground-based coherent DWL). Thus, 2-µm DWL winds are even available for low scattering ratios (<1.1), where no significant aerosol-contamination of the A2D Rayleigh winds can be expected. But we agree with the referee, that the comparison of A2D Rayleigh winds with the 2-µm DWL is limited to atmospheric altitudes, where at least weak aerosol backscattering occurs.

Furthermore, with a view to the Aeolus mission, it is important to note that the strategy for vertical sampling differ between the A2D and the satellite instrument ALADIN. The latter will measure wind profiles from ground up to about 20-30 km altitude, so that the range gates covering the troposphere will generally be fewer and larger compared to the A2D where all the atmospheric range gates are available to sample the altitude range from ground up to about 9 km. For the flights discussed in the manuscript, the vertical sampling grid was chosen such that the wind-shear in the jet stream region could be determined with the highest possible resolution. Hence, the A2D vertical sampling was adapted to the expected wind variability (from short-range NWP forecasts) and science objectives of the flights, which will not be possible for Aeolus, where only a climatology-based approach for different vertical sampling schemes can be applied.

The height assignment error of the Mie channel, comprehensively discussed in (Sun et al., 2014), is thus less pronounced, as the bin height is only 300 m compared to 500 m or 1000 m which would be a typical bin size for the satellite instrument. According to (Sun et al., 2014), the wind error standard deviation grows linearly with increasing bin size. Apart from that, the Rayleigh channel of the A2D exhibits large systematic and random errors owing to the co-alignment issue which has only insignificant influence on the Mie winds. Hence, in terms of the airborne demonstrator, the Mie channel is characterized by higher accuracy and precision compared to the Rayleigh channel. For the satellite instrument, this difference in the performance of both channels is not expected, due to the coarser resolution in the troposphere and the absence of the co-alignment loop. In this sense, we agree with the referee's comment that the remark on page 15: "Mie wind is preferred due to the generally lower systematic and random error" is too strong. A more differentiated discussion of the Rayleigh and Mie wind errors considering the above arguments and how they translate to Aeolus will be included in the revised manuscript.

Changes to the manuscript #1.2:

A new section 4.1.6 was added to the manuscript discussing the Rayleigh and Mie wind errors of the A2D and how the different error sources relate to Aeolus. Also, the statement "the Mie wind is preferred due to the generally lower systematic and random error" was paraphrased to "the Mie wind is preferred due to the higher accuracy and precision of the Mie channel for the A2D".

Changes according to the minor comments of referee #1:

*Page 2:*            *Replace "as it will close the gaps in the wind data coverage" by "as it will contribute to close the gap in wind profile data coverage"*

Response:          The sentence was changed accordingly in the revised version.

*Page 2, line 19:*   *"aircraft" => aircrafts*

Response:          Aircraft is the correct English plural of aircraft.

*Page 5, line 14:*   *Replace "from moving particles (cloud particles, aerosols, molecules)" by "from particles (cloud droplets, aerosols) and molecules which move with the ambient wind"*

Response:          The sentence was changed accordingly in the revised version.

*Page 6:*            $x_0$, $\Delta x$ *and* $k$ *have not been clearly defined near equation 2. Please do.*

Response:          The quantities are now clearly defined in the revised manuscript as follows: "In Eq. (2), $x_0$ represents the Mie fringe centroid position at the frequency $f_0$ of the emitted laser pulse and is referred to as Mie centre. $\Delta x$ is the shift of the Mie fringe centroid position with respect to the Mie centre and $k$ denotes the proportionality factor between the Doppler frequency shift $\Delta f_{Doppler}$ and the resulting shift of the Mie fringe $\Delta x$, thus describing the sensitivity of the Mie channel."

*Caption of Fig. 2:* *"the respective transmitted intensities". Should "transmitted" here not be replaced by "received"?*

Response:          The term "transmitted intensities" refers to the transmission of the light through the two sequential Fabry-Pérot interferometers and therefore appears appropriate from our point of view. For the sake of clarity, "transmitted intensities" was replaced by "intensities transmitted through the filters".

*Caption of Fig. 4:*     *I do not understand the last sentence: "Orange bins are identified as ground bins and thus considered for the determination of the ground response function." Which orange bins? Please explain or correct.*

Response:     In a previous version of the manuscript, the ground bins in Figure 4(c) and (d) were indicated in orange. The colour of the bins was changed to white without adapting the figure caption. The caption was corrected accordingly in the revised version.

**Response to Referee Comment 2**

Comment #2.1:

*Page 3, line 30: Given that other characteristics of the transmitter are stated, it would be useful to know the bandwidth or frequency uncertainty of the transmitted pulse.*

Response to Comment #2.1:

The bandwidth of the transmitted laser pulse (50 MHz) and the frequency stability ($\approx$ 3 MHz rms, UV) will be mentioned in the revised manuscript. More detailed information on the frequency stability of the laser transmitter is provided in Lemmerz et al. (2017).

Changes to the manuscript #2.1:

The following sentence was added to section 2.1: "Concerning the spectral characteristics, the bandwidth of the transmitted UV laser pulses is 50 MHz (FWHM), while the pulse-to-pulse frequency stability is approximately 3 MHz (rms)."

Comment #2.2:

*Page 4, line 3: Why are the receiver electronics triggered 60 microseconds before laser pulse emission?*

Response to Comment #2.2:

The long lead time of the detector electronics is due to an electronic preconditioning process of the accumulating charged coupled device (ACCD) arrays which require a trigger signal being provided 61.4 µs prior to each laser pulse acquisition. The same ACCDs are used for the satellite instrument, but here the long round-trip laser pulse travel time from the satellite to the first atmospheric range gate ($\approx$ 2.5 ms) do not cause an issue from to the long preconditioning process.

Changes to the manuscript #2.2:

The following sentences were added to section 2.1: "The long lead time of the detector electronics is due to an electronic preconditioning process of the accumulating charged coupled device (ACCD) arrays described in section 2.2. Although ACCDs of the same type are used for the satellite instrument, the preconditioning process is not an issue here, since the round-trip laser pulse travel time from the satellite to the first atmospheric range gate ($\approx 2.5$ ms) is sufficiently long."

Comment #2.3:

*Page 4, line 25: It wasn't clear to me exactly what the EOM does. Does it simply switch between the internal reference and the atmospheric signal after the pulse has been transmitted?*

Response to Comment #2.3:

The EOM in the front optics is used to avoid saturation of the ACCD by blocking the atmospheric path for several µs after transmission of the laser pulse, thus preventing strong backscattered light produced close to the instrument (up to about 1 km) from being incident on the detectors. In this way, the EOM separates the internal reference signal, which is guided to the receiver via a multimode fiber and registered in range gate #4, from the atmospheric signal, which enters the receiver on a free-space path including the EOM and is registered in range gates #5 to #24. The EOM is specific to the airborne instrument and not needed on the satellite.

Changes to the manuscript #2.3:

The description of the EOM in section 2.1 was extended as follows: "The EOM is used to avoid saturation of the ACCD by shutting the atmospheric path for several µs after transmission of the laser pulse, thus preventing strong backscattered light produced close to the instrument (up to about 1 km) from being incident on the detectors. In this way, the EOM temporally separates the atmospheric signal from the internal reference signal."

Comment #2.4:

*Page 8, line 18: It would be useful to know why the response function for the satellite instrument is only performed for a single range gate. How is that gate determined? Is the satellite attitude varied so that instrument looks vertically?*

Response to Comment #2.4:

For the satellite instrument the atmospheric Rayleigh response function is derived after adding the return signals obtained from a number of range gates in the upper troposphere (e.g. between 6 km and 16 km) in order to increase the signal-to-noise ratio. The selection of the appropriate range for

averaging is performed during on-ground processing and the information for each single range gate is still included in the downlinked raw data. In the satellite wind retrieval for the L2B product, a Rayleigh-Brillouin line shape model is used in combination with atmospheric temperature and pressure profiles from a NWP model (e.g. from ECWMF) to account for the altitude-dependence of the Rayleigh response over the entire vertical measurement range from ground to the lower stratosphere (Dabas et al., 2008). Hence, in contrast to the A2D, only one set of Rayleigh response calibration parameters is determined for a large vertical range covering multiple range gates. For the A2D a vertical profile of Rayleigh response parameters can be determined, because the SNR is sufficiently high for the airborne instrument. In this regard, the sentence on page 8 "For the satellite instrument, the response function is derived for only one atmospheric range gate" is indeed misleading and will be paraphrased accordingly in the revised manuscript.

As for the second question, the satellite instrument, like the A2D, is operated in nadir-pointing geometry during the response calibration procedure. For this purpose, the whole satellite is rolled by 35°.

Changes to the manuscript #2.4:

The approach for deriving the altitude-dependent Rayleigh response function for the satellite instrument is elaborated in section 3.1 of the revised manuscript.

Comment #2.5:

*Page 9, line 20: I found this paragraph a bit confusing. What does "a visual inspection of intensities" mean, and why does a summation lead to an underestimation of the actual ground signal? Because the paper places quite a bit of emphasis on the improvement of the new technique for dealing with ground return, I think a better characterization of the old technique is necessary.*

Response to Comment #2.5:

The old ground detection scheme was based on an analysis of the curtain plot depicting the Rayleigh and Mie signal intensities after range-correction and normalization to the integration time of each range gate, as shown for the response calibration #6 in Fig. 4(a) and (b). Here, high signal intensities related to strong ground return become visible as white bins, as the intensity exceeds the maximum of the respective colour scale. For this particular example, range gates #21 to #23 would have been (subjectively) selected as ground range gates in the old scheme (by visual inspection by an experienced data analyst), since most of the white bins are found in these three range gates. Only ground signals contained therein would be summed up. However, the new ground detection method based on the signal gradient approach reveals that ground signals are also contained in adjacent

range gates, as displayed in the Rayleigh and Mie ground masks in Fig. 4(c) and (d). For instance, the Mie ground mask features two white bins in range gate #20. In other calibrations or wind scenes, the number of disregarded ground bins can be much larger. Thus, the old ground detection scheme generally involves an underestimation of the actual ground signal, unless a very ample selection of ground range gates is performed. However, the latter approach would significantly increase the atmospheric contribution to the summed ground signal, especially in case of varying ground elevation during the investigated response calibration or wind scene. It should also be noted, that in previous airborne campaigns, the old ground detection scheme was acceptable, since the response calibrations were performed over flat terrain, e.g. sea ice, so that ground signals were almost completely contained in only one range gate.

Changes to the manuscript #2.5:

The above explanations are included in section 3.2 of the revised manuscript.

Comment #2.6:

*Page 10, line 31: I assume that delta h is computed from the DEM data for Table 2, although it isn't totally clear to me from the text.*

Response to Comment #2.6:

Correct. As described on page 9, lines 15ff., $\Delta H$ is the "height difference between a reference ground elevation during one measurement and the upper bin border of the highest (or first) range gate that contains ground signals […]. The reference ground elevation per measurement is derived from the digital elevation model (DEM) ACE2, providing elevation data at a resolution of 9 arc seconds (300 m x 300 m at the equator) (Berry et al., 2010)."

Changes to the manuscript #2.6:

The definition of $\Delta H$ was already included in the original manuscript.

Comment #2.7:

*Page 16, line 25: Because the coherent Doppler lidar measures return from aerosols, and the Rayleigh channel is sensitive to the presence of aerosols, is a comparison between the two valid? Doesn't this potentially overestimate the error in the Rayleigh winds, unless the effect of aerosol on that channel is negligible? Perhaps this should be discussed some more.*

Response to Comment #2.7:

We agree with the referee, that the comparison of A2D Rayleigh winds with the 2-µm DWL is limited to atmospheric regions, where cloud and aerosol backscattering occurs. However, it should be noted that the coherent 2-µm DWL is very sensitive even to weak particulate backscatter return, due to its coherent detection principle with small bandwidth. In addition, since the coherent DWL is deployed on the aircraft, the atmospheric altitudes with low aerosol backscatter are located in near range gates, which do not suffer from the $R^2$-dependency of the signal and strong aerosol extinction (as it would be the case for ground-based coherent DWL). Hence, 2-µm DWL winds are even available for low scattering ratios (<1.1), where no significant aerosol-contamination of the A2D Rayleigh winds can be expected. Furthermore, as described in section 4.1.2, Mie-contaminated bins in the Rayleigh data are identified by a signal threshold approach and excluded from the Rayleigh wind curtain. Such range bins thus do not enter the statistical comparison with the 2-µm DWL winds. Also, the Rayleigh winds are only considered as valid (and enter the statistical comparison with the coherent DWL) in case that no valid winds are detected from the A2D Mie channel by using a SNR threshold on the Mie channel. Thus, we agree that valid A2D Rayleigh winds could be contaminated by narrowband aerosol backscatter, but we consider this effect as not dominating the systematic and random error.

Changes to the manuscript #2.7:

This comment is in line with Comment #1.2. A new section 4.1.6 was added to the manuscript discussing the Rayleigh and Mie wind errors of the A2D including the validity of the comparison of A2D Rayleigh and 2-µm DWL winds.

Comment #2.8:

*Page 17, line 1: The removal of the "bad" measurement from the Rayleigh dataset is done based on comparison with the coherent lidar. How would this be handled for the satellite measurement, where no comparison source will be available, to avoid sending bad data to the assimilation algorithms?*

Response to Comment #2.8:

The removal of "outliers" from the comparison dataset is performed to obtain a probability density function of wind speed differences (Fig. 9 (b) and (c)) which is closer to a Gaussian distribution. In this way, the standard deviation represents a good measure of the random error. However, it is certainly necessary to state the number or percentage of outliers removed from the dataset. We consider "outliers" or "gross errors" as being uniformly distributed over the wind speed

measurement range which add to the Gaussian-distributed random errors. Indeed, the error model for Aeolus, as described in the Mission requirement document MRD (ESA, 2016) does separate these two different errors and defines a requirement for Aeolus on the probability of gross outliers ($< 5\%$). Hence, we think that this approach for a statistical comparison is justified. For the satellite L2B data products, the estimation of the random error is provided for each observation and could be used as additional QC parameter. In addition, NWP centres usually apply a QC (or even variational QC) during the assimilation of the wind products by comparing it with best guess values (background) from the model.

Changes to the manuscript #2.8:

The above explanations have been added to section 4.1.5 of the revised manuscript.

Comment #2.9:

*Page 18, line: 23: Why exactly does a heterogeneous cloud structure contribute to random error? Is it a range-weighting effect in the presence of shear, or some other optical effect? Probably a bit more explanation is needed here.*

Response to Comment #2.9:

Yes, it is the combination of the presence of a strong backscatter gradient (e.g. cloud boundaries) and strong wind shear which is not resolved due to the coarse vertical resolution of A2D and Aeolus. It could be also referred as a Mie height assignment error. As stated in the text, the impact of heterogeneous cloud structure on the Mie random error is comprehensively explained in Sun et al. (2014). The authors found that assigning the measured wind to the centre of the measurement bin introduces an error when particles and/or molecules are not uniformly distributed inside the bin, which is generally the case. Hence, the error originates from a range-weighting effect and is particularly large in the presence of shear in the sampling volume. Assuming a constant wind-shear with typical amplitude of 0.01 s$^{-1}$ over the bin, the random error of the Mie wind scales inversely proportional with the thickness of a particle layer randomly positioned inside the bin, reaching 2 m·s$^{-1}$ for a bin size of 1000 m and a layer thickness of 300 m.

Changes to the manuscript #2.9:

The following sentences were added to the discussion of the Mie height assignment error in the new section 4.1.6: "This so-called height assignment error is especially large in the presence of strong wind shear in the sampling volume. Assuming a constant shear with typical amplitude of 0.01 s$^{-1}$ over the bin, the Mie wind random error scales inversely proportional with the thickness of a

particle layer randomly positioned inside the bin, reaching 2 m·s⁻¹ for a bin size of 1000 m and a layer thickness of 300 m (Sun et al., 2014)."

Comment #2.10:

*Figure 4: The caption refers to "orange bins" in (c) and (d). I'm not sure what bins this refers to; I don't see any orange bins.*

Response to Comment #2.10:

See minor comments of referee #1

Changes to the manuscript #2.10:

See minor comments of referee #1

**Additional changes and corrections**

Changes to the manuscript #3.1:

The axis ticks of the colour scale in Fig. 4(a) and (b) indicating the signal intensities on measurement level have been corrected in order to be consistent with the signal intensities on observation level shown in Fig. 6(a) and (b). Also, the units of the gradient thresholds have been added in section 3.2: $T_{GR,high}$ = 0.015 a.u./km and $T_{GR,low}$ = -0.015 a.u./km (a.u. = arbitrary units).

Changes to the manuscript #3.2:

The unit of the geopotential height in Figs. 5 and 10 has been corrected from "meters" to "dekameters". Moreover, the date in the caption of Fig. 10(b) has been corrected.

Changes to the manuscript #3.3:

The statistical parameters given in Table 3 were calculated after excluding calibration #1, as described in the Table caption. The previous values also considered the $\Delta H$ values, slopes and intercepts obtained from calibration #1, in contradiction with the caption.

Changes to the manuscript #3.4:

The references (Schäfler et al., 2018) and (Reitebuch et al., 2018) were updated.
Reference (Marksteiner et al., 2018) was removed, as the manuscript is still in preparation.

Changes to the manuscript #3.5:

The abbreviation "a.u." was changed to "arb. units".

[revised manuscript text omitted]
$ | $-0.49\ \mathrm{m\cdot s^{-1}}$ | $-0.03\ \mathrm{m\cdot s^{-1}}$ | $-0.21\ \mathrm{m\cdot s^{-1}}$ |
| Mean bias | $-0.49\ \mathrm{m\cdot s^{-1}}$ | $-0.03\ \mathrm{m\cdot s^{-1}}$ | $-0.21\ \mathrm{m\cdot s^{-1}}$ |
| Standard deviation | $2.7\ \mathrm{m\cdot s^{-1}}$ | $1.5\ \mathrm{m\cdot s^{-1}}$ | $2.0\ \mathrm{m\cdot s^{-1}}$ |
| $1.4826 \cdot$ median absolute deviation | $2.6\ \mathrm{m\cdot s^{-1}}$ | $1.3\ \mathrm{m\cdot s^{-1}}$ | $1.8\ \mathrm{m\cdot s^{-1}}$ |

5  **Table 5.** Results of the statistical comparison between the A2D and the 2-μm LOS wind data measured on 04/10/2016. The statistical comparison for the Mie wind profiles was performed without and with ZWC.

| Statistical parameter | Rayleigh winds | Mie winds (without ZWC) | Mie winds (with ZWC) |
|---|---|---|---|
| Number of compared bins | 168 | 1246 | 1246 |
| Number of removed bins due to gross error ($>\pm10\ \mathrm{m\cdot s^{-1}}$) | 11 | 0 | 0 |
| Correlation coefficient $r$ | 0.96 | 0.99 | 0.99 |
| Slope $A$ | $1.01 \pm 0.02$ | $1.04 \pm 0.03$ | $1.04 \pm 0.03$ |
| Intercept $B$ | $1.67\ \mathrm{m\cdot s^{-1}}$ | $0.55\ \mathrm{m\cdot s^{-1}}$ | $0.02\ \mathrm{m\cdot s^{-1}}$ |
| Mean bias | $1.54\ \mathrm{m\cdot s^{-1}}$ | $0.57\ \mathrm{m\cdot s^{-1}}$ | $0.04\ \mathrm{m\cdot s^{-1}}$ |
| Standard deviation | $3.3\ \mathrm{m\cdot s^{-1}}$ | $1.9\ \mathrm{m\cdot s^{-1}}$ | $1.9\ \mathrm{m\cdot s^{-1}}$ |
| $1.4826 \cdot$ median absolute deviation | $2.7\ \mathrm{m\cdot s^{-1}}$ | $2.0\ \mathrm{m\cdot s^{-1}}$ | $2.0\ \mathrm{m\cdot s^{-1}}$ |